# Allelic variation of *TaWD40-4B.1* contributes to drought tolerance by modulating catalase activity in wheat

Geng Tian[1], Shubin Wang[2], Jianhui Wu [3], Yanxia Wang[4], Xiutang Wang[4], Shuwei Liu[1], Dejun Han[3], Guangmin Xia [1] ✉ & Mengcheng Wang [1] ✉

Drought drastically restricts wheat production, so to dissect allelic variations of drought tolerant genes without imposing trade-offs between tolerance and yield is essential to cope with the circumstance. Here, we identify a drought tolerant WD40 protein encoding gene *TaWD40-4B.1* of wheat via the genome-wide association study. The full-length allele $TaWD40\text{-}4B.1^C$ but not the truncated allele $TaWD40\text{-}4B.1^T$ possessing a nonsense nucleotide variation enhances drought tolerance and grain yield of wheat under drought. TaWD40-4B.1$^C$ interacts with canonical catalases, promotes their oligomerization and activities, and reduces $H_2O_2$ levels under drought. The knock-down of catalase genes erases the role of $TaWD40\text{-}4B.1^C$ in drought tolerance. $TaWD40\text{-}4B.1^C$ proportion in wheat accessions is negatively correlative with the annual rainfall, suggesting this allele may be selected during wheat breeding. The introgression of $TaWD40\text{-}4B.1^C$ enhances drought tolerance of the cultivar harboring $TaWD40\text{-}4B.1^T$. Therefore, $TaWD40\text{-}4B.1^C$ could be useful for molecular breeding of drought tolerant wheat.

Drought is one of the main abiotic stresses and poignantly affects crop growth and yield. Wheat (*Triticum aestivum* L.), a staple crop, is mainly cultivated in arid and semi-arid regions worldwide, so its yield is frequently compromised by water scarcity[1]. This circumstance is aggravated by the increasing water resource deficiency. It is therefore very crucial and urgent to build drought-tolerant germplasms and cultivars without yield penalty via efficient strategies such as molecular assistant breeding[2]. This requires a comprehensive dissection of the genetic basis of drought tolerance, with the aim to mine excellent genes that directly and efficiently target the principal physiological basis of drought response and have no negative impact on growth and yield.

Drought tolerance is a complex trait controlled by many genes with minor effects. In the past several decades, some quantitative trait loci (QTLs) of drought tolerance have been mined through the linkage analysis of segregation populations[3,4]. Genome-wide association study (GWAS) is a powerful tool for dissecting the genetic basis of complicated agronomic traits. It is especially efficient for mining the allelic variations that have been selected during domestication and breeding, which appears to avoid the trade-offs between tolerance and yield, and therefore has promising potential in germplasm improvement. Given genomic complexity and a high proportion of repetitive sequences, the application of GWAS in wheat lags behind other major crops such as rice and maize[5–9]. Up to now, all of the QTLs controlling drought tolerance in wheat were present as genetic positions, except for a NAC transcription factor *TaNAC071-A* that was recently identified via GWAS[10]. Apart from those regulating transcriptional levels, the allelic variations causing the alteration of peptide sequences may be the other evolutionary force affecting phenotype and trait improvement, but this kind of allelic variations in

[1]The Key Laboratory of Plant Development and Environment Adaptation Biology, Ministry of Education, School of Life Science, Shandong University, 266237 Qingdao, Shandong, P. R. China. [2]Institute of Vegetable Research, Shandong Academy of Agricultural Sciences, 250100 Jinan, Shandong, P. R. China. [3]State Key Laboratory of Crop Stress Biology for Arid Areas, College of Agronomy, Northwest A&F University, 712100 Yangling, Shaanxi, P. R. China. [4]Shijiazhuang Academy of Agriculture and Forestry Sciences, 050050 Shijiazhuang, Hebei, P. R. China. ✉e-mail: xiagm@sdu.edu.cn; wangmc@sdu.edu.cn

drought-tolerant genes have been rarely identified via GWAS and other genetic analyses so far. On the other hand, a series of genetic analyses, derived from segregation populations and GWAS in the past fifteen years, identified a consistent major genomic region *qDSI.4B.1* on the short arm of chromosome 4BS (see Fig S3 and the references therein). This region is responsible for up to 22% of phenotypic variation under drought stress[11]. Several consistent meta-QTLs locate in this interval[12], one of which, the major-effect MQTL$_{1.4}$ is also present in chromosome 4BS of tetraploid durum wheat as well as drought tolerance QTL regions of maize and barley based on the syntenic analysis[13], making this location even more interesting and important for further scrutiny[14]. As yet, the drought-tolerant allelic gene in this locus has not been cloned.

As a principal physiological response to drought and other abiotic stress, excessive accumulation of reactive oxygen species (ROS), including $H_2O_2$, causes oxidative damage in plants[15–17]. On the other hand, $H_2O_2$ serves as a signaling molecule for modulating the signaling transduction network to enhance abiotic stress tolerance and modulate plant growth[18,19]. Plants have evolved types of efficient regulatory machinery including ROS scavengers to fine-tune ROS homeostasis and avoid toxic ROS levels for adapting to abiotic stresses[18]. The transcriptional regulation of the activities of scavenging enzymes during the response to abiotic stress has been widely investigated[20–23], but the modulatory scenario at the post-transcriptional level is still not well addressed in plants. It has been found that catalase, the important $H_2O_2$ scavenger, can be regulated by interacting with chaperones such as AtNCA1, AtLSD1, small heat shock protein AtHsp17.6CII, natriuretic peptide AtPNP-A in Arabidopsis, and ROD1, NCA1 isoforms, UDP-xylose synthase OsUXS3 and SRL10 in rice, and Alpha-momorcharin (αMMC) in *Momordica charantia*[24–32], and through post-translational modifications such as phosphorylation by OsSTRK1, AtCPK8 and AtSOS2 and ubiquitination by OsAPIP6[33–36] in the response to abiotic and biotic stresses. The activity of catalase can also be directly modulated by the effectors such as AvrPiz-t, CMV2b, and triple gene block protein 1 (TGBp1) from pathogens[24,37–39]. In animal cells, the common mechanism for activating catalase is proved to form oligomers (mostly tetramers) with high activity, while monomeric catalases have no or low activity[40]. The oligomerization of catalases has been seldom reported in plants; a study in rice found that catalases form dodecameric holoenzymes containing 12 subunits, and holoenzyme has high catalase activity and a fast turnover rate for $H_2O_2$[41]. However, the factors (chaperones) modulate the oligomerization and therefore the activity of catalases has not been identified in plants.

WD40 domain-containing proteins (WD40) constitute a large gene family among eukaryote species[42] and usually serve as the scaffolds and chaperones to regulate protein interaction and function[43]. In plants, WD40 genes have been reported to modulate various biological processes, such as the grain yield of maize and rice[44,45] and the adaptation to drought and other abiotic stress[46,47], but the underlying mechanism has not been well unraveled. Whether WD40 proteins directly target the principal physiological responses upon drought and other abiotic stresses such as post-transcriptionally modulating catalase and other ROS scavenging enzymes is unknown so far.

In this work, we conduct a GWAS and identify a drought-tolerant WD40 gene *TaWD40-4B.1* embedding in *qDSI.4B.1* locus on chromosome 4BS of wheat. The allelic nonsense variation of *TaWD40-4B.1* produces the allele *TaWD40-4B.1$^T$* encoding a truncated peptide which is associated with lower drought tolerance in the wheat accessions. The complete allele *TaWD40-4B.1$^C$* enhances drought tolerance and grain yield under drought stress but does not impose their trade-offs. TaWD40-4B.1$^C$ but not TaWD40-4B.1$^T$ strongly interacts with canonical catalases and promotes their oligomerization and activity under drought stress. Thus, our research reveals that the *TaWD40-4B.1$^C$* allele plays a significant role in wheat drought tolerance.

## Results

### GWAS for wheat drought tolerance at the seedling stage

In order to mine drought tolerance-associated genes, a diverse panel of 198 common wheat natural accessions representing the diversity of genetic structure and geographical distribution of the wheat population was used to evaluate the genetic basis controlling drought tolerance (Supplementary Data 1 and 2). These accessions were genotyped by the 660 K single-nucleotide polymorphism (SNP) genotyping array. A total of 419,606 high-quality SNP markers were retained for analysis after filtering (MAF < 0.05 and missing data > 0.1), and these SNP markers were located on 21 chromosomes (Supplementary Fig. 1a). These accessions were clustered into six subpopulations by population structure and kinship analyses based on these filtered SNPs (Supplementary Fig. 1b–d and Supplementary Data 2). Then, we phenotyped the drought tolerance of these accessions at the seedling stage using the drought tolerance (DT) index, which was defined as the grades ranging from one to six according to the wilting extent of the first to the fourth leaves of the seedlings (Supplementary Fig. 2a–c). The selected accessions exhibited diverse drought tolerance with a large variation in DT indices ranging from one to six (Supplementary Fig. 2d and Supplementary Data 1).

To dissect the genetic loci governing drought tolerance, we performed a GWAS based on the DT index dataset using the filtered SNPs. Through a Bonferroni-adjusted correction, we set the *P*-value for the suggestive threshold as $2.383 \times 10^{-6}$. Under the mixed linear model (MLM) with correction of the population structure (Q) and kinship (K), 55 SNPs were identified to be significantly associated with drought tolerance (Fig. 1a, b), and they were distributed in four linkage disequilibrium (LD) blocks on chromosomes 4A, 4B, 7B, and 4D. Of them, the leading SNP, AX-109453581, at the position of 67352769 on the short arm of chromosome 4B (4BS) was identified ($P = 4.181 \times 10^{-8}$; Phenotypic variance explanation = 14.34%; Fig. 1a, b and Supplementary Data 3). The LD block on 4BS is embedded in the drought-tolerant locus *qDSI.4B.1* that has been widely reported in the past 15 years (refer to Supplementary Fig. 3 for examples and references). The LD block containing the leading SNP harbored 12 genes (Fig. 1c and Supplementary Data 4), whose expression profiles were evacuated among 48 randomly selected wheat accessions with DT indices ranging from 1.2 to 3 as well as from 5 to 6 (24 for each) (Supplementary Data 5). Five genes were transcribed under the control and/or drought stress, three of which (*TraesCS4B02G072100*, *TraesCS4B02G072200*, *TraesCS4B02G072300*) were induced by drought stress, and *TraesCS4B02G072200* showed the highest transcriptional level under the drought condition (Supplementary Figs. 2 and 4a). However, the change folds of their expression upon drought stress were comparable between the accessions with low (1.2–3) and high (5–6) DT indices (Supplementary Fig. 4b–f), indicating that drought tolerance is not associated with the transcriptional regulation of these genes in the LD block.

The leading SNP mentioned above was located at the 14th nucleotide of *TraesCS4B02G072200*'s 3′-UTR (Fig. 1d). *TraesCS4B02G072200* is an intron-less gene and encodes a typic WD40-repeat protein containing seven WD40 domains, along with two adjacent paralogs *TraesCS4B02G072100* and *TraesCS4B02G072300* to form a WD40 gene tandem cluster (Supplementary Fig. 5), coinciding with the phenomenon that some of the WD40 genes are distinguished as tandem duplication in wheat[48]. Besides the leading SNP at 3′-UTR, three SNPs in the coding sequence (CDS) of *TraesCS4B02G072200* were excavated based on Sanger sequencing of 198 accessions, and they were all linked with the leading SNP (Fig. 1d and Supplementary Data 6). Of these four linked SNPs, SNP465 and SNP999 were synonymous, while SNP1274 (G → A) was nonsense and caused the substitution of Trp codon TGG to stop codon TAG (Fig. 1e). Thus, the allelic variations in CDS of *TraesCS4B02G072200* resulted in two haplotypes, one encoding a complete peptide and the other encoding a truncated

peptide (Fig. 1h). The truncated peptide had lower prediction confidence of the seventh WD40 domain, and its 3D structure was affected (Supplementary Fig. 6). The seedlings of wheat accessions possessing

complete haplotype exhibited higher DT indices than those with truncated haplotype (Fig. 1f). Hence, *TraesCS4B02G072200* was the most putative candidate drought-tolerant gene and was named

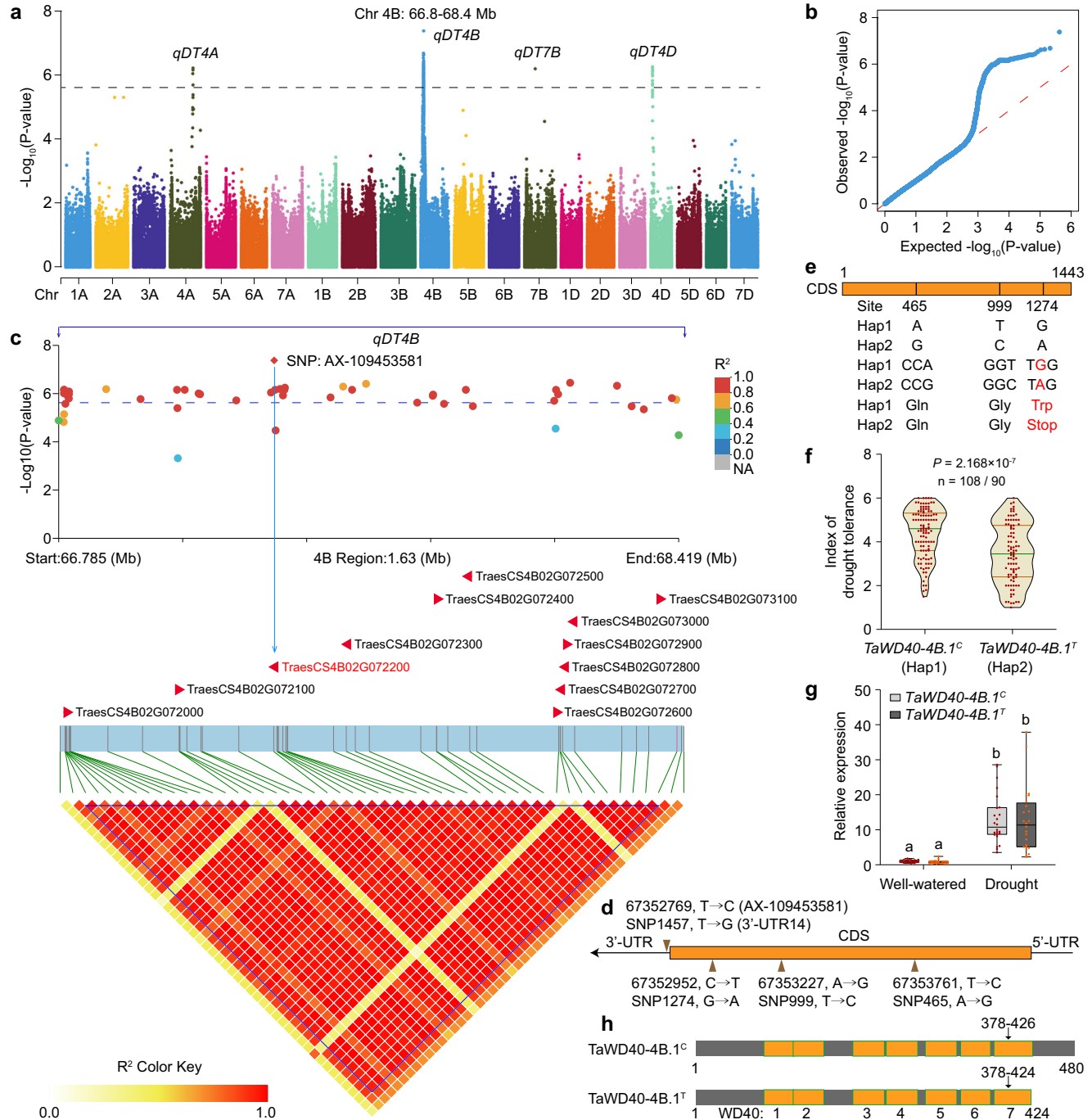

**Fig. 1 | The allelic variation of *TaWD40-4B.1* is associated with drought tolerance in wheat. a** Results of the GWAS for wheat drought tolerance. The horizontal dashed line represents the genome-wide suggestive significance threshold ($P = 2.383 \times 10^{-6}$). **b** Quantile–quantile plot for the GWAS under the mixed linear model (MLM). **c** The 12 genes and association analysis between genetic variations and drought tolerance within the LD block on chromosome 4B. *TaWD40-4B.1* is labeled in red. Blue line indicates the candidate region (66.785–68.419 Mb) for significant marker–trait associations on chromosome 4B. **d** The scheme of four absolutely linked SPNs in the positions in the genomic sequence of chromosome 4B and *TaWD40-4B.1* gene. AX-109453581 is the number of SNP marker that exhibited the leading significant association with DT index. **e** Four SNPs in the coding sequence and 3'-UTR of *TaWD40-4B.1* causing complete (*TaWD40-4B.1^C*) and

truncated (*TaWD40-4B.1^T*) haplotypes. **f** The comparison of drought tolerance (DT) index between the accessions harboring *TaWD40-4B.1^C* and *TaWD40-4B.1^T* haplotypes using the two-sided z-test (n = 108 and 90 wheat accessions; $P = 2.168 \times 10^{-7}$). Brown lines, upper and lower quartiles; green lines, medians. **g** The expression of *TaWD40-4B.1* in the accessions harboring *TaWD40-4B.1^C* and *TaWD40-4B.1^T* haplotypes under the well-watered and water-withheld conditions (n = 24 wheat accessions; $P = 1.14 \times 10^{-14}$). Box indicates the range from lower to upper quartiles, and bar ranges the minimum to maximum observations. The significance of the difference is calculated with a one-way ANOVA analysis–Tukey comparison and the columns labeled without the same alphabet are significantly different ($P < 0.05$). **h** The diagram indicates the length and WD40 domains of TaWD40-4B.1^C and TaWD40-4B.1^T. Source data are provided as a Source data file.

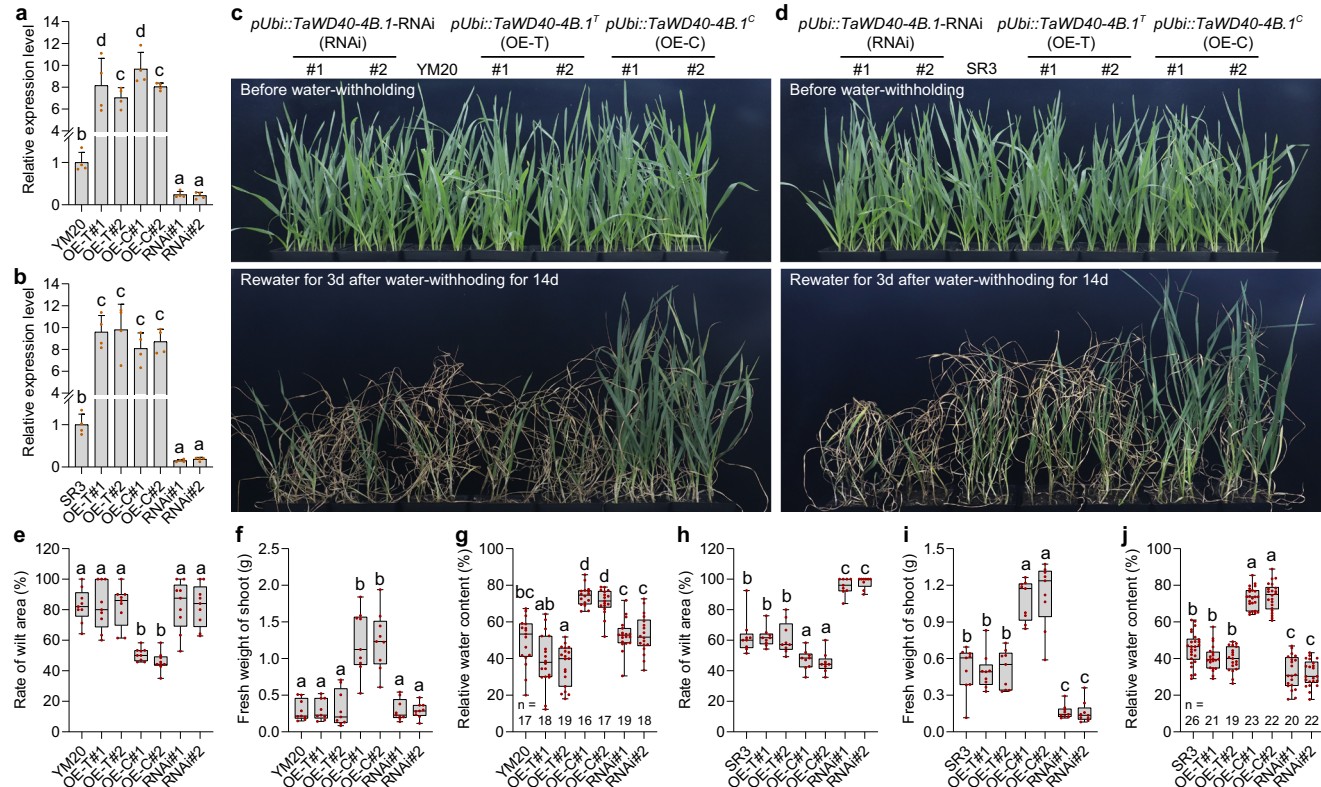

**Fig. 2 | *TaWD40-4B.1^C* but not *TaWD40-4B.1^T* enhances drought tolerance of wheat seedlings. a, b** The expression level of *TaWD40-4B.1^C* and *TaWD40-4B.1^T* in transgenic lines of cultivars YM20 (**a**) and SR3 (**b**) (*n* = 4 biologically independent samples; *P* = 1.45 × 10⁻¹⁴ in **a** and 1.07 × 10⁻¹¹ in **b**). **c, d** The water-withholding assay of the transgenic lines of YM20 (**c**) and SR3 (**d**). **e, h** The rates of wilting area in the leaves of wheat seedlings after water-withholding (*n* = 9 biologically independent samples; *P* = 5.26 × 10⁻¹¹ in **e** and 1.08 × 10⁻²³ in **h**). **f, i** The fresh weight of shoots of wheat seedlings after water-withholding (*n* = 9 biologically independent samples; *P* = 4.29 × 10⁻¹⁵ in **f** and 8.09 × 10⁻²¹ in **i**). **g, j** The relative water content of wheat seedlings after water-withholding (*n* present in the panels; *P* = 9.11 × 10⁻²³ in **g** and

5.87 × 10⁻⁵⁵ in **j**). In **a, b**, data are shown as mean and standard deviation; In **e–j**, the box indicates the range from lower to upper quartiles, and the bar ranges the minimum to maximum observations. The significance of the difference is calculated with a one-way ANOVA analysis–Tukey comparison and the columns labeled without the same alphabet are significantly different (*P* < 0.05, two-sided). SR3 and YM20: two cultivars carrying *TaWD40-4B.1^C* and *TaWD40-4B.1^T*, respectively; OE-T: *TaWD40-4B.1^T* overexpression lines; OE-C: *TaWD40-4B.1^C* overexpression lines; RNAi: *TaWD40-4B.1^C* / *TaWD40-4B.1^T* RNAi lines. Source data are provided as a Source data file.

*TaWD40-4B.1*. In comparison with the accessions with complete haplotype (*TaWD40-4B.1^C*), those with truncate haplotype (*TaWD40-4B.1^T*) produced comparable transcripts of *TaWD40-4B.1* under both well-watered and water-limited conditions (Fig. 1g). Meanwhile, the expression levels of *TaWD40-4B.1*'s paralogs as well as the other two transcriptionally detectable genes in the LD block were also equivalently similar between the accessions with *TaWD40-4B.1^C* and *TaWD40-4B.1^T* alleles (Supplementary Fig. 4g–j), indicating that the allelic variation is not associated with the transcriptional regulation of these genes. In line with these results, we concluded that *TaWD40-4B.1* was the candidate gene of the drought-tolerant locus on chromosome 4BS, and its complete and truncated alleles were associated with the variation in drought tolerance of the wheat accessions.

### *TaWD40-4B.1^C* but not *TaWD40-4B.1^T* enhanced the drought tolerance of wheat seedlings

To confirm the role of *TaWD40-4B.1^C* and *TaWD40-4B.1^T* in drought tolerance, we selected two cultivars, Yangmai 20 (YM20) carrying the *TaWD40-4B.1^T* allele and Shanrong 3 (SR3) carrying the *TaWD40-4B.1^C* allele respectively, to construct transgenic lines. Of them, both *TaWD40-4B.1^C* and *TaWD40-4B.1^T* overexpression lines (labeled as OE-C and OE-T, respectively) of either YM20 or SR3 were transformed with *TaWD40-4B.1^C* and *TaWD40-4B.1^T* driven by the ubiquitin promoter (pUbi) respectively, and their knock-down lines were generated via the RNA interference (RNAi) strategy (labeled as RNAi lines). To avoid the

influence of the transcript dosage effect, the overexpression (OE) lines that produced similar transcripts of *TaWD40-4B.1^C* and *TaWD40-4B.1^T* were selected for water-withholding assay (Fig. 2a, b). In these RNAi lines, the expression levels of *TraesCS4B02G072100* and *TraesCS4B02G072300*, two paralogs of *TaWD40-4B.1*, were comparable to those of the wild-type (Supplementary Fig. 7), showing that RNAi did not affect the transcription of the paralogs.

Under the well-watered condition, there was no morphological difference between YM20 or SR3 and their corresponding *TaWD40-4B.1^C*/*TaWD40-4B.1^T* overexpression and RNAi lines (Fig. 2c, d). Drought stress stimulated by water withholding caused the seedlings to wilt (Fig. 2c, d). Both YM20 and the OE-T lines were wilted with the same strength, and their shoot fresh weights and relative water contents were significantly reduced; while in comparison with YM20 and the OE-T lines, the OE-C lines exhibited superior growth capacity, the leaf wilting was attenuated, and the fresh weights and relative water contents were higher (Fig. 2c, e–g). Consistently, *TaWD40-4B.1^C* overexpression enhanced drought tolerance, alleviated leaf wilting, and the reduction of shoot fresh weight and relative water content in SR3 after water-withholding, while *TaWD40-4B.1^T* overexpression had no visible effect (Fig. 2d, h–j). On the other hand, the knock-down of *TaWD40-4B.1^C* in SR3 reduced drought tolerance, while the knock-down of *TaWD40-4B.1^T* in YM20 did not visibly alter the tolerance (Fig. 2c–j). Collectively, these data indicate that the *TaWD40-4B.1^C* but not *TaWD40-4B.1^T* improves the drought tolerance of wheat.

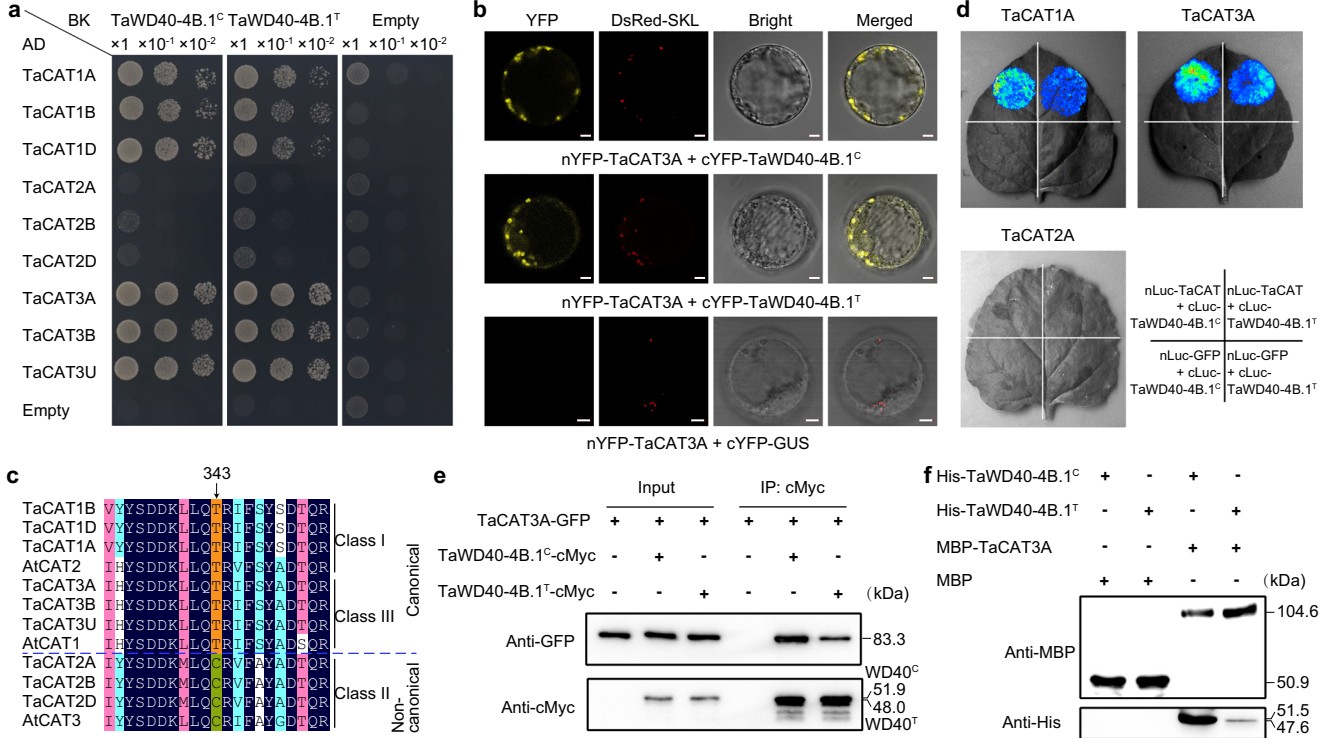

**Fig. 3 | TaWD40-4B.1$^C$ has a stronger interaction capacity with canonical catalases than TaWD40-4B.1$^T$. a** The yeast two hybridization assay reveals that both TaWD40-4B.1$^C$ and TaWD40-4B.1$^T$ interact with canonical TaCAT1s and TaCAT3s but not noncanonical TaCAT2s. **b** The bimolecular fluorescence complementation assay confirms the interaction between TaWD40-4B.1$^C$/TaWD40-4B.1$^T$ and TaCAT3A. DsRed-SKL is the DsRed fused with peroxisome targeting sequence SKL serving as the peroxisome marker. Bar = 5 μm. **c** TaCAT1s/TaCAT3s and TaCAT2s are canonical and non-canonical catalases, respectively. **d** The split-luciferase complementation imaging showing TaWD40-4B.1$^C$ has a stronger interaction capacity with TaCAT1A and TaCAT3A than TaWD40-4B.1$^T$. **e, f** Co-immunoprecipitation (**e**) and pull-down (**f**) assays indicate that TaWD40-4B.1$^C$ has a stronger interaction capacity with TaCAT3A than TaWD40-4B.1$^T$. Source data are provided as a Source data file.

## TaWD40-4B.1$^C$ had stronger interaction with canonical catalases than TaWD40-4B.1$^T$

To explore the mechanism of *TaWD40-4B.1*-mediated drought tolerance, we identified potential TaWD40-4B.1-interacting proteins through screening a yeast two-hybridization library of wheat. Among these, two proteins annotated as catalase 3A (TaCAT3A) and TaCAT3U were identified repeatedly. There have three catalase genes (*TaCAT1*, *TaCAT2*, and *TaCAT3*) in the wheat genome, and each of them has three alleles located in A, B, and D subgenomes (Supplementary Fig. 8a). CATs have three classes in plants; according to the phylogeny and synteny relationships between TaCATs and CAT orthologues of other plants, TaCAT1, TaCAT2, and TaCAT3 are categorized into classes I, III, and II, respectively (Supplementary Fig. 8b), and are concurrently clustered with orthologous AtCAT2, AtCAT3 and AtCAT1 of Arabidopsis[49,50]. The yeast two-hybridization assay showed that both TaWD40-4B.1$^C$ and TaWD40-4B.1$^T$ interacted with TaCAT1s and TaCAT3s but not TaCAT2s (Fig. 3a). We confirmed the interaction by the bimolecular fluorescence complementation assay (BiFC) using the A subgenome alleles of TaCAT1, TaCAT2, and TaCAT3 because the alleles among three subgenomes share high identity (Supplementary Fig. 8a). The result also shows that TaWD40-4B.1$^C$ and TaWD40-4B.1$^T$ interacted with TaCAT1A and TaCAT3A but not TaCAT2A via the transient co-expression of N-terminal yellow fluorescence protein (nYFP) fused TaCAT and cYFP fused TaWD40-4B.1 in wheat protoplasts (Fig. 3b and Supplementary Fig. 9). These findings indicate that TaWD40-4B.1 specifically interacts with TaCATs in classes I and III rather than class II. Recently, AtCAT3 (Class II) was verified to be a non-canonical catalase that functions as a transnitrosylase, whose activity depended on the unique Cys-343 residue that is highly conserved in plants, while AtCAT1 and AtCAT2 (Classes III and I, respectively) were

canonical catalases characterized as the Thr-343 residue[51]. Interestingly, three alleles (TaCAT2A, TaCAT2B, and TaCAT2D) of class II TaCATs carry the conserved Cys-343 residue, while class I and III TaCATs carry Thr-343 (Fig. 3c and Supplementary Fig. 8b). These findings indicate that both TaWD40-4B.1$^C$ and TaWD40-4B.1$^T$ interacts with canonical catalases in class I and III rather than non-canonical catalases in class II in wheat.

Transient co-expression of GFP-fused proteins and peroxisome marker DsRed-SKL (SKL is a peroxisome targeting sequence) in wheat protoplasts showed that GFP-TaWD40-4B.1$^C$ and GFP-TaWD40-4B.1$^T$ did not localize in the peroxisomes but dispersed in the cells as GFP alone (Supplementary Fig. 10a). Alike rice CATs[41], canonical TaCAT1 and TaCAT3 possess a peroxisomal targeting signal (PTS1) with the signaling motif (SRL) localized at −7 to −9 upstream of the C terminus, but noncanonical TaCAT2A has no PTS1 (Supplementary Fig. 8c); GFP-TaCAT1A and GFP-TaCAT3A majorly accumulated close to the peroxisomes, but GFP-TaCAT2A dispersed in the cells with no obvious fluorescence signal around the peroxisomes (Supplementary Fig. 10). The place of TaWD40-4B.1 and TaCATs in cells was further confirmed by the western blot assay using the total, cytoplasmic and peroxisomal proteins extracted from the protoplasts (Supplementary Fig. 10b). In wheat protoplasts, the YFP fluorescence signals of the interaction between TaWD40-4B.1$^C$/TaWD40-4B.1$^T$ and canonical TaCATs accumulated close to the peroxisomes (Fig. 3b and Supplementary Fig. 9).

Given that the truncation was predicted to affect the structure of TaWD40-4B.1 (Supplementary Fig. 6 and Supplementary Data 7), we compared the interaction capacity of TaWD40-4B.1$^C$ and TaWD40-4B.1$^T$ with TaCATs using the split-luciferase complementation imaging (SLCI) assay (Fig. 3d). Both TaWD40-4B.1$^C$ and TaWD40-4B.1$^T$ interacted with TaCAT1A and TaCAT3A rather than TaCAT2A in *Nicotiana*

*benthamiana* leaves, further confirming the specific interaction between TaWD40-4B.1 and canonical catalases. Moreover, the fluorescence signal densities produced by the interaction of TaWD40-4B.1[C] and TaCAT1A/TaCAT3A were stronger than those between TaWD40-4B.1[T] and TaCAT1A/3 A, showing TaWD40-4B.1[C] had a higher affinity with canonical TaCATs. Because the expression of *TaCAT3s* instead of *TaCAT1s* was increased under drought stress[49], which indicates the close association between TaCAT3 and drought response, we chose TaCAT3A to further confirm the difference in interaction affinity. Firstly, TaCAT3A fused with GFP (TaCAT3A-GFP) and TaWD40-4B.1[C] fused with cMyc tag (TaWD40-4B.1[C]-cMyc)/TaWD40-4B.1[T]-cMyc were transiently co-expressed in *N. benthamiana* leaves to conduct the co-immunoprecipitation assay. TaCAT3A-GFP was precipitated by either TaWD40-4B.1[C]-cMyc or TaWD40-4B.1[T]-cMyc, but the abundance of TaCAT3A-GFP precipitated by TaWD40-4B.1[C]-cMyc was higher than that by TaWD40-4B.1[T]-cMyc (Fig. 3e). Moreover, His-TaWD40-4B.1[C]/ His-TaWD40-4B.1[T] and MBP-TaCAT3A being expressed, respectively, in *E. coli* were purified for the pull-down assay. After in vitro incubation, His-TaWD40-4B.1[C]/His-TaWD40-4B.1[T] were both pulled down by MBP-TaCAT3A, but not MBP alone, and pulled-down His-TaWD40-4B.1[C] had a higher abundance than His-TaWD40-4B.1[T] (Fig. 3f). These data demonstrate that TaWD40-4B.1[C] has a stronger interaction capacity with canonical TaCATs than TaWD40-4B.1[T] does.

## TaWD40-4B.1[C] but not TaWD40-4B.1[T] promoted in vitro and in vivo catalase activity

Given that TaWD40-4B.1[C] was more prone to interact with canonical TaCATs than TaWD40-4B.1[T], we compared the effects of the two haplotypes on catalase activity. First, we found that both the overexpression and RNAi lines of *TaWD40-4B.1[C]/TaWD40-4B.1[T]* had comparable expression levels of *TaCAT1* and *TaCAT3* to either SR3 or YM20 under either well-watered or drought conditions (Supplementary Fig. 11), showing the genetic manipulation of *TaWD40-4B.1* did not affect the expression of catalase genes. The lines of *TaWD40-4B.1[C]* and *TaWD40-4B.1[T]* overexpression and *TaWD40-4B.1[C]* RNAi possessed comparable catalase activities to SR3 in the whole leaves of seedlings under the well-watered condition (Fig. 4a). After exposure to drought stress, the catalase activities were elevated; in comparison with SR3, *TaWD40-4B.1[C]* overexpressors had higher activities, its knock-down lines lower, but *TaWD40-4B.1[T]* overexpression lines were comparable. A similar result was obtained in YM20 and its transgenic lines (Supplementary Fig. 12a). Then, we used a yeast fast-heterologous model system to further analyze the effects of TaWD40-4B.1[C] and TaWD40-4B.1[T] on in vivo catalase activity (Fig. 4b–d). The yeast cells expressing either TaWD40-4B.1[C] or TaWD40-4B.1[T] alone possessed catalase activity similar to the control transformed with an empty vector (Fig. 4c), showing recombinant TaWD40-4B.1[C] or TaWD40-4B.1[T] had no effect on the activity of yeast catalase. The expression of TaCAT3A increased the total catalase activity in yeast; in comparison with the yeast cells expressing TaCAT3A alone, those co-expressing TaWD40-4B.1[T] had similar catalase activity, while those co-expressing TaWD40-4B.1[C] had obviously higher catalase activity (Fig. 4c). Consistently, all the recombinants had similar growth rates to the control under the non-stressful condition (Fig. 4d). The exposure to 3 mM $H_2O_2$ slowed down the growth rates of all yeast lines. In comparison with the control, the yeast expressing TaWD40-4B.1[C] or TaWD40-4B.1[T] alone had a comparable growth rate. The yeast producing TaCAT3A exhibited more vigorous growth capacity than the control, in accord with the role of durum wheat TdCAT1[52], and co-expression of TaWD40-4B.1[C] but not TaWD40-4B.1[T] promoted the positive effect of TaCAT3A on the growth ability (Fig. 4d). This indicates the increase in the in vivo catalase activity of TaCAT3A by TaWD40-4B.1[C] enhanced the tolerance to oxidative stress in yeast. Meanwhile, we measured the role of TaWD40-4B.1[C] and TaWD40-4B.1[T] using the in vitro catalase activity assay (Fig. 4e). TaCAT3A alone had quite weak catalase activity and the

presence of TaWD40-4B.1[T] had no distinguishable effect on the activity of TaCAT3A. The application of TaWD40-4B.1[C] triggered the in vitro activity of TaCAT3A, and the elevation of activity became more drastic following the increase of TaWD40-4B.1[C] concentration.

It has proved that oligomerization of catalases is crucial for activating their activities[40], and catalases oligomerize to dodecameric holoenzyme containing 12 units with high activity in rice[41]. First, the SLCI assay revealed that the homo-interaction occurred between TaCAT1A and TaCAT1A as well as between TaCAT3A and TaCAT3A, and the presence of TaWD40-4B.1[C] but not TaWD40-4B.1[T] promoted the homo-interactions (Fig. 4f, g). Then, the proteins of wheat leaves were cross-linked by dithiobis (succinimidyl propionate) (DSP) for immunoassay to detect the effect of TaWD40-4B.1 on the oligomerization of catalases using the antibody of TaCAT3 (56.4 kDa). Under both well-watered and drought conditions, in comparison with SR3, the abundances of dimeric and oligomeric catalases were increased by the overexpression of *TaWD40-4B.1[C]* rather than *TaWD40-4B.1[T]*, but decreased by the RNAi of *TaWD40-4B.1[C]* (Fig. 4h). When the cross-links between proteins were cleaved by β-mercaptoethanol (β-ME) to destroy the protein-protein interaction, only monomeric catalases were detected with similar abundances among SR3 and the transgenic lines under either well-watered or drought condition (Fig. 4h), showing that TaWD40-4B.1 does not affect the expression of catalases. We further analyzed the role of TaWD40-4B.1 in the oligomerization of TaCAT3A in the reaction system of in vitro catalase activity assay using the same methodology (Fig. 4i). In the absence of TaWD40-4B.1, the MBP-fused TaCAT3A (104.6 kDa) was not oligomerized. The addition of TaWD40-4B.1[C] induced the oligomerization of TaCAT3A, and the abundance of TaCAT3A oligomers became more pronounced with the rise of TaWD40-4B.1[C] concentration, while the promotion to TaCAT3A oligomerization caused by TaWD40-4B.1[T] turned out to be negligible. Different from the in vivo assay, dimerized TaCAT3A could hardly be detected in the in vitro system. These data indicate that TaWD40-4B.1[C] rather than TaWD40-4B.1[T] contributes more to the oligomerizations of canonical catalases and therefore increases their activities in wheat.

## *TaWD40-4B.1[C]* enhanced the tolerance to oxidative stress in wheat

Given that *TaWD40-4B.1[C]* enhanced catalase activity, we further analyzed the role of *TaWD40-4B.1[C]* and *TaWD40-4B.1[T]* in the tolerance of wheat seedlings to oxidative stress simulated via supplying exogenous $H_2O_2$ in a liquid medium (Fig. 5a, b). In the absence of $H_2O_2$ treatment, the seedlings of SR3 and its transgenic lines were comparable to each other. When supplied with 100 mM $H_2O_2$, the growth of SR3 seedlings was affected, and the plant height was reduced. *TaWD40-4B.1[C]* overexpression alleviated the growth restriction of SR3, and its knock-down aggravated the effect of oxidative stress; while *TaWD40-4B.1[T]* overexpression had no obvious performance. We then measured the $H_2O_2$ level in the leaves of wheat seedlings planted in soil using the DAB staining assay (Fig. 5c). Under the well-watered condition, the $H_2O_2$ levels were similar among SR3 and all lines. Under drought stress, the $H_2O_2$ levels were increased; in comparison with SR3, the increase was weaker in *TaWD40-4B.1[C]* overexpression lines, more significant in the knock-down lines, but similar in the *TaWD40-4B.1[T]* overexpression lines. Consistently, the contents of malonaldehyde (MDA) representing the extent of in vivo oxidative damage were comparable among all plants under the well-watered condition (Fig. 5d). After exposure to drought stress, the MDA contents were increased significantly, and *TaWD40-4B.1[C]* overexpression attenuated the accumulation of MDA, its knock-down had the opposite effect, while *TaWD40-4B.1[T]* overexpression did not influence the MDA contents. The parallel analysis using YM20 and its *TaWD40-4B.1[C]/TaWD40-4B.1[T]* overexpression lines confirmed the role of *TaWD40-4B.1[C]* and *TaWD40-4B.1[T]* on oxidative stress and $H_2O_2$ levels as well (Supplementary Fig. 12b–e). Because ROS are often produced under all kinds of abiotic stresses, we measured the

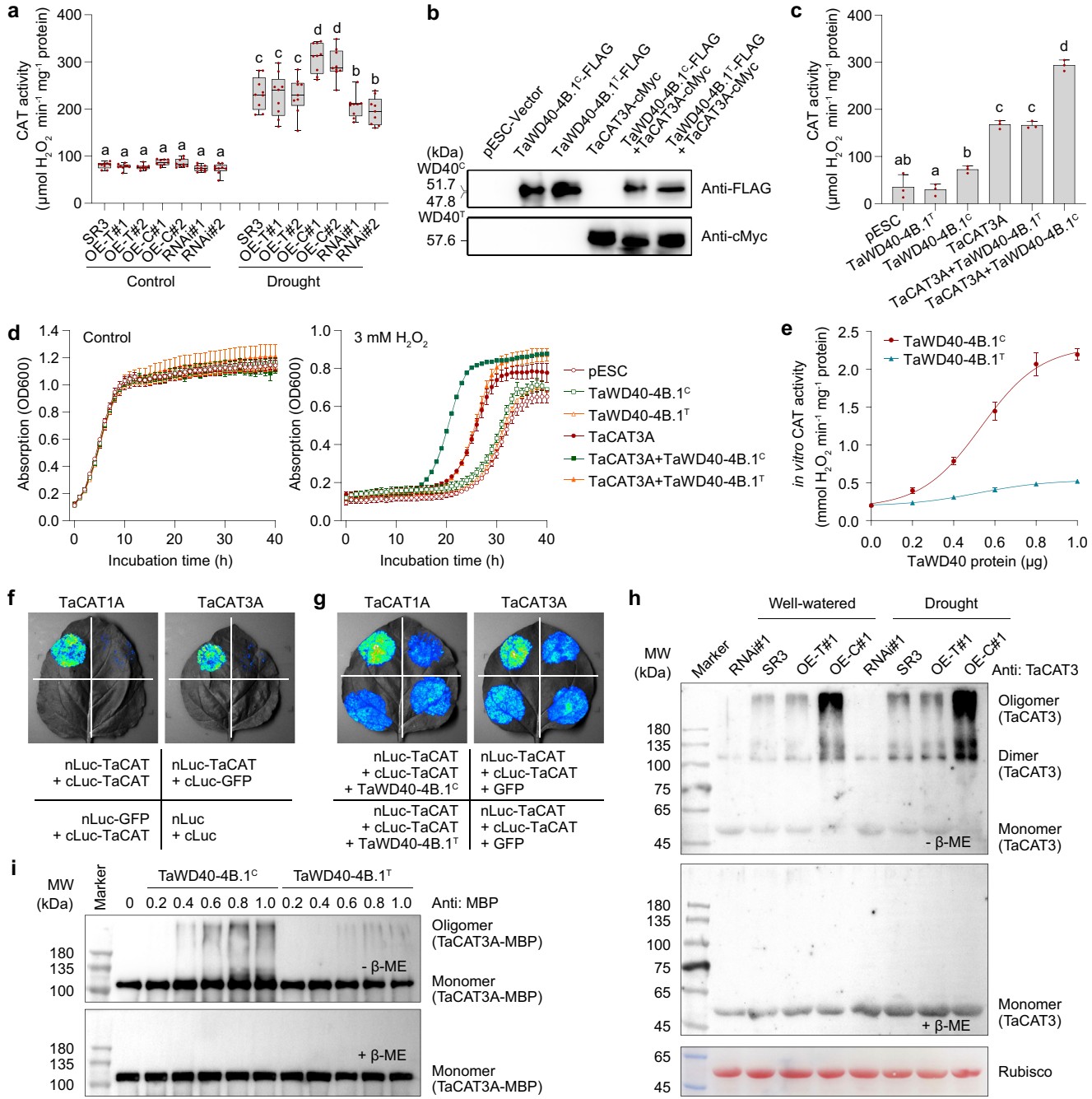

**Fig. 4 | TaWD40-4B.1$^C$ but not TaWD40-4B.1$^T$ promotes in vivo and in vitro activity and oligomerization of catalase. a** The catalase activities in the leaves of cultivar SR3 and its transgenic lines under the well-watered and water-withheld conditions ($n = 9$ biologically independent samples; $P = 6.36 \times 10^{-59}$). Box indicates the range from lower to upper quartiles, and the bar ranges the minimum to maximum observations. SR3: the cultivar carrying *TaWD40-4B.1$^C$*; OE-T: *TaWD40-4B.1$^T$* overexpression lines; OE-C: *TaWD40-4B.1$^C$* overexpression lines; RNAi: *TaWD40-4B.1$^C$* RNAi lines. **b** The ectopic expression of TaWD40-4B.1$^C$/TaWD40-4B.1$^T$ and/or TaCAT3A in yeast confirmed by the western blotting assay. **c** The catalase activities of the yeast cells ectopically expressing TaWD40-4B.1$^C$/TaWD40-4B.1$^T$ and/or TaCAT3A ($n = 3$ biologically independent samples; $P = 1.60 \times 10^{-10}$). **d** The growth rates of the yeast cells ectopically expressing TaWD40-4B.1$^C$/TaWD40-4B.1$^T$ and/or TaCAT3A under the control and H$_2$O$_2$-treated conditions

($n = 3$ biologically independent samples). **e** TaWD40-4B.1$^C$ promotes in vitro activity of TaCAT3A ($n = 5$ biologically independent samples). **f** TaCAT1A and TaCAT3A interact with themselves. **g** TaWD40-4B.1$^C$ promotes the interaction of TaCAT1A-TaCAT1A and TaCAT3A-TaCAT3A. **h** The monomer, dimer, and oligomer of catalases are detected in the cross-linked proteins by DSP extracted from wheat seedlings, but the dimer and oligomer are not detected in the proteins when the cross-links are cleaved by β-ME. **i** The oligomerization of catalases is promoted by TaWD40-4B.1$^C$ in vitro. The assay is the same as **h** using the proteins from in vitro catalase activity assay system. In **c**–**e**, data are shown as mean and standard deviation. In **a**, **c**, the significance of the difference is calculated with a one-way ANOVA analysis–Tukey comparison, and the columns labeled without the same alphabet are significantly different ($P < 0.05$, two-sided). Source data are provided as a Source data file.

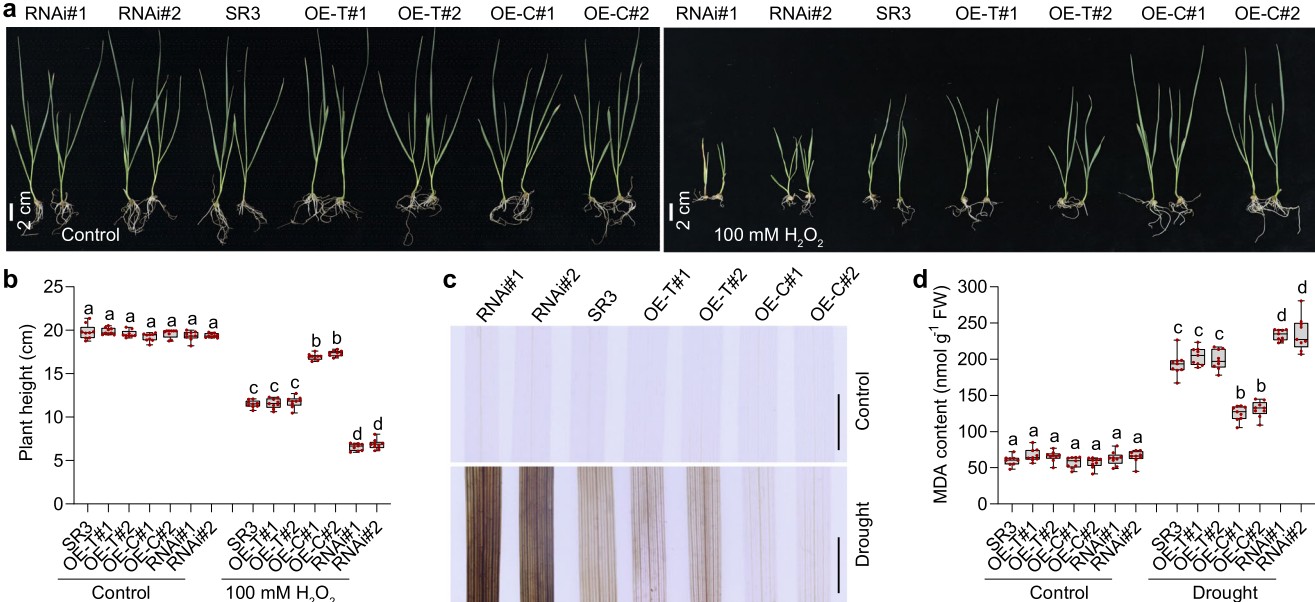

**Fig. 5 | *TaWD40-4B.1^C* but not *TaWD40-4B.1^T* enhances tolerance to oxidative stress and reduces H₂O₂ level under drought stress. a** The assessment of the oxidative stress tolerance of SR3 and its transgenic lines. Bar = 2 cm. **b** The statistical result of plant height in **a** ($n = 9$ biologically independent samples; $P = 1.30 \times 10^{-106}$). **c** The H₂O₂ levels revealed by the DAB staining in the leaves of wheat seedlings under the well-watered and water-withheld conditions. Bar = 1 cm. **d** The MDA contents in the leaves of wheat seedlings under the well-watered and water-withheld conditions ($n = 9$ biologically independent samples; $P = 1.58 \times 10^{-84}$).

In **b–d**, the box indicates the range from lower to upper quartiles, and the bar ranges the minimum to maximum observations; the significance of the difference is calculated with a one-way ANOVA analysis–Tukey comparison and the columns labeled without the same alphabet are significantly different ($P < 0.05$, two-sided). SR3: the cultivar carrying *TaWD40-4B.1^C*; OE-T: *TaWD40-4B.1^T* overexpression lines; OE-C: *TaWD40-4B.1^C* overexpression lines; RNAi: *TaWD40-4B.1^C* RNAi lines. Source data are provided as a Source data file.

role of *TaWD40-4B.1* in the tolerance to salt, alkali, and heat, the major abiotic stresses during the life course of wheat (Supplementary Fig. 13). *TaWD40-4B.1* that constitutively expressed at the young and reproductive stages was induced by dehydration in leaves but not roots (Supplementary Fig. 13a, b). Thus, we measured the expression profiles of *TaWD40-4B.1* upon these stresses and H₂O₂ treatment in leaves, and found that *TaWD40-4B.1* was also induced by the stimuli (Supplementary Fig. 13c). In comparison with SR3, the *TaWD40-4B.1^C* overexpression and RNAi lines had higher and lower tolerance respectively to salt, alkali, and heat stress, while the *TaWD40-4B.1^T* overexpressors had similar tolerance capacities (Supplementary Fig. 13d–f). These data demonstrate that *TaWD40-4B.1^C* can avoid the over-accumulation of H₂O₂ (ROS) to enhance drought tolerance.

## *TaWD40-4B.1^C* enhanced drought tolerance via catalase to avoid ROS over-accumulation

To confirm whether the role of *TaWD40-4B.1^C* in drought tolerance is due to promoting catalase activity, we conducted a genetic analysis using the *TaCAT* knock-down (RNAi) lines of SR3 that the expression of canonical catalase genes *TaCAT1* and *TaCAT3* was obviously reduced (Supplementary Fig. 14a). *TaCAT* knock-down slightly restricted the growth under the well-watered condition (alike *CATB* mutation in rice[24]), and promoted the leaf wilting when the seedlings suffered from drought stress (Fig. 6a). The RNAi line had lower catalase activity than SR3, and the difference was more significant under drought (Fig. 6e), showing the decrease in catalase activity by *TaCAT* knock-down reduced drought tolerance. Then we crossed the *TaCAT* RNAi line and the *TaWD40-4B.1^C*/*TaWD40-4B.1^T* overexpression lines to construct the *TaWD40-4B.1* genetic lines in the catalase knock-down background (Supplementary Fig. 14b–d). Under the well-watered condition, the seedlings of the cross lines were slightly shorter than those of the *TaWD40-4B.1^C*/*TaWD40-4B.1^T* overexpression lines (Fig. 6d). After exposure to drought stress, as

observed above, the *TaWD40-4B.1^C* overexpression line exhibited superior drought tolerance than SR3 and the *TaWD40-4B.1^T* overexpression line; knock-down of *TaCATs* reduced the drought tolerance of SR3 and its *TaWD40-4B.1* transgenic lines, and almost abolished the positive role of *TaWD40-4B.1^C* in drought tolerance (Fig. 6b–d). Consistently, the role of *TaWD40-4B.1^C* in promoting catalase activity under drought was also lost when *TaCATs* were knocked down (Fig. 6e). High light intensity induces photorespiration, which leads to the generation of large amounts of H₂O₂ in the peroxisomes, resulting in severe plant growth retardation[53]. Catalase is a direct target of hydroxyurea, and hydroxyurea inhibits catalase activity[54]. Under the high light intensity, SR3 had superior growth capacity than the *TaWD40-4B.1^C* RNAi lines in the absence of hydroxyurea, whilst the growth difference became weaker in the presence of low concentration (1 mM) of hydroxyurea, and comparable when applied high concentration (2 and 3 mM) of hydroxyurea (Supplementary Fig. 15). On the other hand, we randomly selected 24 wheat accessions harboring *TaWD40-4B.1^C* and 24 wheat accessions harboring *TaWD40-4B.1^C* to compare their catalase activities (Supplementary Data 5), and found that the accessions harboring *TaWD40-4B.1^C* had higher catalase activities than those with *TaWD40-4B.1^T* under drought (Fig. 6f). These data indicate the contribution of *TaWD40-4B.1^C* to drought tolerance is, at least in large part, achieved through its direct promotion of catalase activity to reduce ROS over-accumulation.

## *TaWD40-4B.1^C* overexpression enhanced grain yield under water-withheld conditions

As *TaWD40-4B.1^C* enhanced drought tolerance at the seedling stage, we further analyzed its role in grain yield by the water-withholding assay with plants at the reproductive stage grown in pots. Under the well-watered condition, SR3 and its transgenic lines showed similar growth ability during the whole life cycle, while under the water-

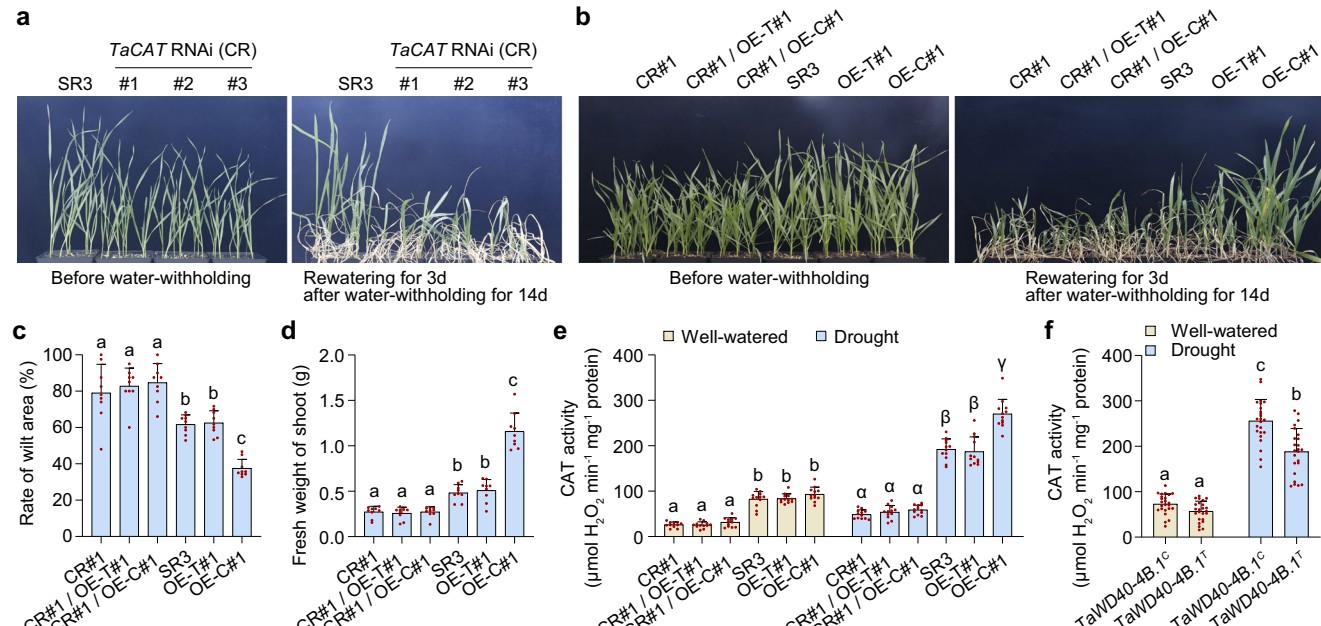

**Fig. 6 | The role of *TaWD40-4B.1^C* in drought tolerance is dependent on catalases. a** The knock-down of *TaCATs* reduces the drought tolerance of wheat. CR#1-3: three *TaCAT* RNAi lines of SR3 carrying *TaWD40-4B.1^C*. **b** The knock-down of *TaCATs* abolishes the role of *TaWD40-4B.1^C* in drought tolerance. CR#1/OE-T#1: *TaCAT* RNAi/*TaWD40-4B.1^T* overexpression cross line; CR#1/OE-C#1: *TaCAT* RNAi/*TaWD40-4B.1^C* overexpression cross line. **c**, **d** the rate of wilt area (**c**) and fresh weight (**d**) of the leaves after drought as shown in **b** (*n* = 9 biologically independent samples; *P* = 4.30 × 10^−14 in **c** and 7.85 × 10^−23 in **d**). **e** The catalase activities in the leaves (*n* = 12 biologically independent samples; *P* = 1.14 × 10^−28 under well-watered condition and 4.26 × 10^−39 under drought). **f** The catalase activity in the wheat accessions harboring *TaWD40-4B.1^T* and *TaWD40-4B.1^C*, respectively (*n* = 24 wheat accessions; *P* = 3.15 × 10^−35). In **c**–**f**, data are shown as mean and standard deviation, the significance of the difference is calculated with a one-way ANOVA analysis–Tukey comparison, and the columns labeled without the same alphabet are significantly different (*P* < 0.05, two-sided). Source data are provided as a Source data file.

withheld condition, in comparison with SR3, *TaWD40-4B.1^C* overexpression lines had a less wilting extent, *TaWD40-4B.1^C* knock-down lines exhibited an opposite phenotype and *TaWD40-4B.1^T* overexpression lines had no obvious difference (Fig. 7a), consistent with the difference in drought tolerance at the seedling stage. The grain sizes were comparable under the well-watered condition; water-withholding reduced the grain sizes, in comparison with SR3, the grain sizes of *TaWD40-4B.1^C* overexpression and knock-down lines became larger and smaller respectively, but those of *TaWD40-4B.1^T* overexpression lines were comparable (Fig. 7b–d). Consistently, 1000-grain weights (TGWs) were similar to each other under the well-watered condition, and water-withheld treatment reduced TGWs and differentiated them among the lines, with the highest in *TaWD40-4B.1^C* overexpression lines, the lowest in the knock-down lines (Fig. 7e). The other yield-associated indices including spike number, grain number per ear, and plant height, respectively, had no difference among SR3 and its transgenic lines under both well-watered and water-withheld conditions (Fig. 7f–h). Under the well-watered condition, there was no difference in grain yield; under the water-withheld condition, *TaWD40-4B.1^C* overexpression alleviated the decrease in grain yield and the overexpression lines had higher grain yields than SR3, *TaWD40-4B.1^C* knock-down exhibited an opposite effect, and *TaWD40-4B.1^T* overexpression lines had similar grain yields to SR3 (Fig. 7i). We also measured the effect of *TaWD40-4B.1^C* genetic manipulation on water use efficiency (WUE). SR3 and all the transgenic lines had comparable CO$_2$ assimilation rates, transpiration rates, and WUE under the well-watered condition (Fig. 7j–l). Under the water-withheld condition, in comparison to SR3, *TaWD40-4B.1^C* overexpression lines had a lower transpiration rate but higher CO$_2$ assimilation rate and WUE, its knock-down lines opposite, and these indices of *TaWD40-4B.1^T* overexpression lines were similar. The

parallel yield trial using YM20 and its transgenic lines phenocopied the results of SR3 (Supplementary Fig. 16a–k).

We further compared the yield of YM20 and its transgenic lines in the field with two irrigation conditions (Supplementary Fig. 16l–o). In the field irrigated with 1500 m³ ha⁻¹ of water, spike densities, and grain numbers per spike were comparable among YM20 and all lines, but TGWs of *TaWD40-4B.1^C* overexpression lines were higher (>2%) than those of YM20 and *TaWD40-4B.1^T* overexpression lines; in the field not irrigated with water, *TaWD40-4B.1^C* overexpression lines had similar grain numbers per spike but higher spike densities (~5%) and TGWs (5.5 and 6.0%) than YM20 and *TaWD40-4B.1^T* overexpression lines (Supplementary Fig. 16l–n). Under two irrigation conditions, in comparison with YM20, *TaWD40-4B.1^T* overexpression lines had comparable grain yields, but *TaWD40-4B.1^c* overexpression lines had higher by 6.9–7.9% and 12.9–16.3%, respectively (Supplementary Fig. 16o). In line with these, we found that *TaWD40-4B.1^C* but not *TaWD40-4B.1^T* enhances grain yield under water-withheld conditions.

**The introduction of *TaWD40-4B.1^C* enhanced drought tolerance**
Given the overexpression of *TaWD40-4B.1^C* but not *TaWD40-4B.1^T* enhanced tolerance and yield of wheat under drought stress, we further analyzed whether these two haplotypes could alter drought tolerance each other via introducing *TaWD40-4B.1^C* of drought tolerant cultivar Luyuan301 into drought sensitive cultivar Chuanmai36 harboring *TaWD40-4B.1^T* allele as well as introducing *TaWD40-4B.1^T* of Chuanmai36 into Luyuan301 through successive backcrossing to construct near-isogenic lines (NILs). Under the well-watered condition, there had no difference between Chuanmai36 and its NIL^*TaWD40-4B.1C* as well as Luyuan301 and its NIL^*TaWD40-4B.1T*; after water-withholding, Chuanmai36 exhibited lower drought tolerance than its NIL^*TaWD40-4B.1C*, oppositely Luyuan301 possessed higher drought tolerance than its NIL^*TaWD40-4B.1T* (Fig. 8a, b). This indicates that the introgression of

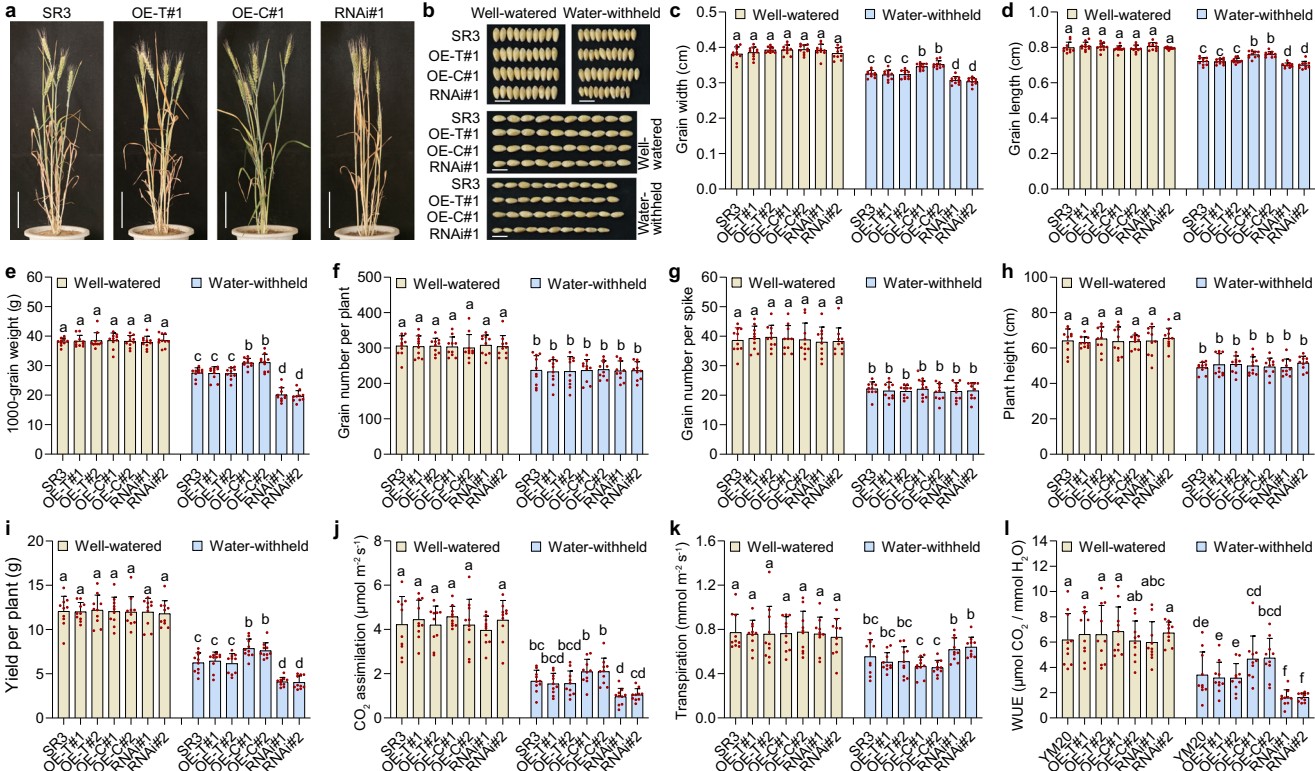

**Fig. 7 | _TaWD40-4B.1^C_ enhances grain yields under the water-withheld condition. a** The representative seedlings under water-withheld condition. Bar = 10 cm. **b** The grain sizes of SR3 and its transgenic lines under the well-watered and water-withheld conditions. Bar = 1 cm. **c**, **d** The statistical comparison of grain width (**c**) and length (**d**) in **b** ($n = 10$ biologically independent samples; $P = 1.48 \times 10^{-51}$ in **c** and $P = 5.13 \times 10^{-42}$ in **d**). **e** The 1000-grain weights under the well-watered and water-withheld conditions ($n = 10$ biologically independent samples; $P = 6.27 \times 10^{-62}$). **f–h** The grain number per plant (**f**), grain number per spike (**g**), and plant height (**h**) under the well-watered and water-withheld conditions ($n = 10$ biologically independent samples; $P = 1.06 \times 10^{-17}$ in **f**, and $P = 3.91 \times 10^{-46}$ in **g** and $P = 1.45 \times 10^{-22}$ in **h**). **i** The grain yield of plants under the well-watered and water-withheld conditions

($n = 10$ biologically independent samples; $P = 8.41 \times 10^{-49}$). **j–l** The $CO_2$ assimilation (**j**), transpiration rate (**k**), and water use efficiency (**l**) under the well-watered and water-withheld conditions ($n = 10$ biologically independent samples; $P = 5.15 \times 10^{-38}$ in **j**, and $P = 6.45 \times 10^{-11}$ in **k**, and $P = 2.17 \times 10^{-19}$ in **l**). In **c–l**, data are shown as mean and standard deviation, the significance of the difference is calculated with a one-way ANOVA analysis–Tukey comparison, and the columns labeled without the same alphabet are significantly different ($P < 0.05$, two-sided). SR3: the cultivar carrying _TaWD40-4B.1^C_; OE-T: _TaWD40-4B.1^T_ overexpression lines; OE-C: _TaWD40-4B.1^C_ overexpression lines; RNAi: _TaWD40-4B.1^C_ RNAi lines. Source data are provided as a Source data file.

_TaWD40-4B.1^C_ indeed enhances drought tolerance, while _TaWD40-4B.1^T_ in replacement of _TaWD40-4B.1^C_ decreases drought tolerance. On the other hand, we analyzed the distribution of two _TaWD40-4B.1_ haplotypes using 600 accessions from different districts in China (Supplementary Data 8). The wheat accessions possessing _TaWD40-4B.1^C_ (80.8%) were more than those carrying _TaWD40-4B.1^T_ (19.2%) (Fig. 8c). Of them, the cultivars and breeding materials had lower proportions of _TaWD40-4B.1^C_, but the landraces had higher. Then, 584 of 600 accessions from 16 provinces/cities (the main districts of wheat production) of China were used to evaluate the relationship between the proportion of _TaWD40-4B.1_ haplotypes and annual rainfall (Supplementary Data 8). There were less accessions carrying _TaWD40-4B.1^T_ in northern districts with lower annual rainfalls but fewer ones in southern districts with higher annual rainfalls, and the proportions of _TaWD40-4B.1^T_ exhibited a close positive correlation with the annual rainfalls (Fig. 8d and Supplementary Data 9). In line with the public SNP information[55–58], we further found that there also had two haplotypes of _TaWD40-4B.1_ in the accessions around the world (Supplementary Fig. 17 and Supplementary Data 10 and 11). In consistency with the accessions in China, the proportions of accessions with _TaWD40-4B.1^C_ were higher than those with _TaWD40-4B.1^T_. Comparatively, the proportions of _TaWD40-4B.1^C_ in Asia and Oceania were higher, but those in America, Africa, and Europe were lower, and the proportion in West Europe was the lowest (Supplementary Fig. 17 and Supplementary Data 11). In comparison with total accessions, the cultivars and

breeding lines had higher _TaWD40-4B.1^T_ proportion, but the old cultivars had lower, close to the proportion of the landraces (Supplementary Fig. 17b).

## Discussion

Natural variation offers the elite agricultural traits during the domestication and breeding of crops. Thus, to mine the elite allelic variations of stress-tolerant genes that do not impose the trade-offs between the yield and tolerance[59] is meaningful for dealing with adverse environmental stimuli. In this study, we performed a GWAS and identified a drought tolerance-associated locus qDT4B with the highest significant confidence on chromosome 4BS (Fig. 1). This locus is embedded in a consistent major genomic region qDSI.4B.1 of drought tolerance in both bread and durum wheat, as well as some other crops, that have been widely reported (see Supplementary Fig. 3 and the references therein), showing it is an important drought-tolerant locus of wheat and other crops. The drought tolerance of qDT4B locus owes to the WD40 gene _TaWD40-4B.1_, because the wheat accessions carrying truncated allele _TaWD40-4B.1^T_ exhibit lower drought tolerance capacity than those with complete allele _TaWD40-4B.1^C_ (Fig. 1), and _TaWD40-4B.1^T_ is deficient in the role of enhancing drought tolerance, water use efficiency, and grain yield under the water-withheld condition that is performed by the _TaWD40-4B.1^C_ (Figs. 2, 7, and 8 and Supplementary Fig. 16). More importantly, both overexpression and knockdown of _TaWD40-4B.1^C_ have no adverse effect on plant growth

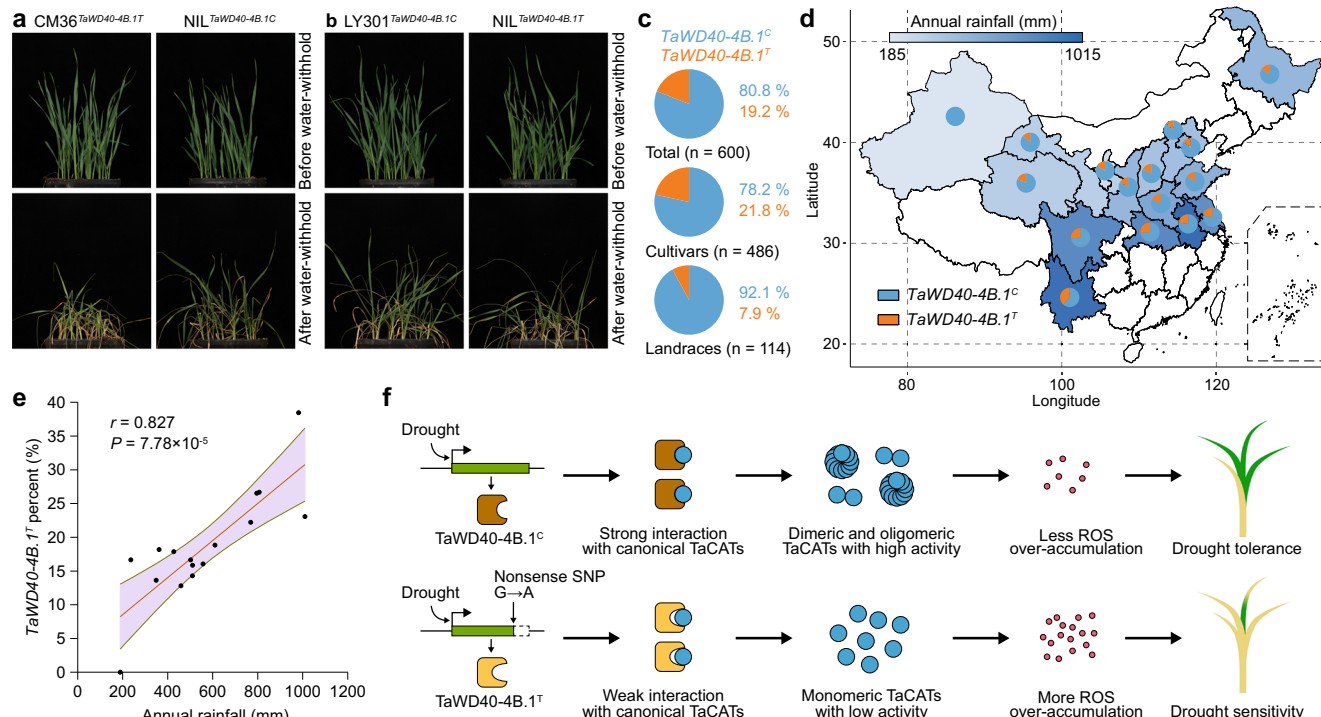

**Fig. 8 | The Introgression of *TaWD40-4B.1^C* enhances drought tolerance and the *TaWD40-4B.1^C* proportion is correlative with annual rainfall. a** Introgression of *TaWD40-4B.1^C* enhances drought tolerance of the accession harboring with *TaWD40-4B.1^T*. **b** Introgression of *TaWD40-4B.1^T* decreases drought tolerance of the accession harboring with *TaWD40-4B.1^C*. **c** The proportions of *TaWD40-4B.1^C* and *TaWD40-4B.1^T* in 600 Chinese wheat accessions. **d** The proportions of *TaWD40-4B.1^C* and *TaWD40-4B.1^T* in the provinces/cities of the major wheat-planted district in China. **e** The proportion of *TaWD40-4B.1^T* is positively correlative with annual rainfall with the Pearson correlation analysis (*n* = 16 wheat planting districts; *P* = 7.78 × 10^-5). **f** The working model of TaWD40-4B.1^C and TaWD40-4B.1^T in drought tolerance of wheat. Source data are provided as a Source data file.

and grain yield under the well-watered condition (Figs. 2, 5, and 7 and Supplementary Fig. 16). This indicates that *TaWD40-4B.1^C* is an elite gene (haplotype) for molecular breeding of drought-tolerant cultivars without imposing the trade-offs between grain yield and drought tolerance and has the potential in water-saving agriculture for coping with the increasing circumstance of global water resource deficiency. Besides, given its contribution to the tolerance to high salt, alkali, and heat (Supplementary Fig. 13), *TaWD40-4B.1^C* appears to have the potential for breeding germplasms with broad-spectrum tolerance to multiple abiotic stress.

As members of a large family, WD40 proteins participate in multiple processes such as development, metabolite biosynthesis, and immune and stress responses[60–62]. It has been reported that WD40 proteins perform modulatory roles by serving as scaffolds for protein-protein interactions[42,63]. For instance, maize WD40 proteins KRN2 and Shrek1 modulate grain development via interacting with a DUF1644-containing protein and regulating pre-rRNA processing, respectively[44,45]; Arabidopsis WD40 protein COP1 regulates photomorphism by cooperating with a series of regulatory factors[64]. WD40 proteins are involved in the MYB-bHLH-WD40 complex to transcriptionally modulate the biosynthesis of anthocyanins and other flavonoids, the secondary metabolites with non-enzymatic antioxidant capacity, which contribute to the tolerance for drought and other abiotic stress via decreasing ROS level[65,66]. In this work, we found *TaWD40-4B.1^C* interacts with canonical catalases and promotes their activity, reduces $H_2O_2$ levels under drought, and the knock-down of catalase genes abolishes the role of *TaWD40-4B.1^C* in drought tolerance (Figs. 3–6 and Supplementary Fig. 12). Catalase, a crucial enzymatic scavenger for removing excessive $H_2O_2$, is closely associated with drought stress[67]. Especially, the catalase coding gene of *CAT2* (grouped in class I canonical catalase) is identified as an eQTL of

drought tolerance in maize based on the eGWAS[68]. Thus, apart from modulating protein-protein interactions such as MYB and bHLH transcriptional factors' interaction, the WD40 protein (TaWD40-4B.1^C) offers drought tolerance by promoting catalase activity to avoid $H_2O_2$ (ROS) over-accumulation, the typical and principal physiological response to adverse environmental stimuli.

Among the catalase-interacting proteins, AtLSD1 interacts with all of the three catalases[29], OsROD1 and AtNCA1 interact with canonical OsCatB and AtCAT2 respectively, but whether they can interact with other canonical and non-canonical catalase (OsCatC/AtCAT1 and OsCatA/AtCAT3, respectively) have not been detected[24,25]. In our research, TaWD40-4B.1 interacts with canonical TaCAT1s and TaCAT3s but not non-canonical TaCAT2s (Fig. 3), and appears to be chaperone specific to canonical catalase in plants, providing further evidence for its crucial contribution to excessive $H_2O_2$ scavenging. The active catalases are matured through heme acquisition and oligomer (majorly tetramer in animal cells) formation and are modulated by phosphorylation and ubiquitination[33,34]. Here, TaWD40-4B.1^C can promote the in vitro activity of catalase (Fig. 4e) in the solution not adding heme and lacking the condition for catalase modification (TaWD40-4B.1 has no property for catalyzing the modification). Thus, TaWD40-4B.1^C promotes catalase activity largely by helping catalases oligomerize (Fig. 4f–i). It has proved that catalases form dodecameric holoenzymes with high catalase activity and fast turnover rate of $H_2O_2$ in rice[41], and the molecular weight of oligomeric TaCATs is larger than that of tetrameric forms (Fig. 4h, i), so it is speculated that TaWD40-4B.1^C can promote the formation of dodecameric catalase holoenzymes and, consequently, enhance catalase activity and lower $H_2O_2$ levels under drought (Figs. 4a and 5 and Supplementary Fig. 12). In line with these data, the maturation of catalases is achieved via forming oligomerized (dodecamerized) holoenzymes instead of tetramers in plant cells,

indicating the distinguishable mechanisms of catalase maturation between plants and animals. TaWD40-4B.1$^C$ is therefore a factor modulating catalase oligomerization and provides a target for further understanding the molecular basis of catalase maturation.

Functional WD40 proteins require at least seven WD40 domains to form the structure of seven-bladed beta-propellers with a funnel-like shape stabilized by the disulfide bond connection between the first and last blades[69]. TaWD40-4B.1$^T$ deficient C-terminus starting from the last bipeptide of the seventh WD40 domain (Fig. 1h) has a slight difference in structure from TaWD40-4B.1$^C$ (Supplementary Fig. 6). Thus, TaWD40-4B.1$^T$ may lose chaperone function, so it can't interact with canonical catalases and promote their oligomerization and activity to enhance drought tolerance (Figs. 2–7). On the other hand, besides as a toxic molecule resulting in oxidative damage when over-accumulation, $H_2O_2$ serves as an important signaling molecule to modulate the signaling transduction networks of abiotic stress response and growth and reproduction. To fine-tune ROS ($H_2O_2$) homeostasis through orchestrating ROS production and/or scavenging machineries will balance plant growth and stress tolerance. TaWD40-4B.1$^C$ does not affect catalase activity, $H_2O_2$ level, growth ability, and grain yield under the well-watered condition (Figs. 4, 5, and 7 and Supplementary Fig. 12). This indicates the complicated modulation mechanisms of TaWD40-4B.1 in catalase activity in planta, and other unknown factors along with TaWD40-4B.1 may orchestrate ROS homeostasis and avoid imposing the trade-offs between grain yield and drought tolerance. Further exploration is needed to uncover the molecular and structural mechanisms underlying the interaction of TaWD40-4B.1$^C$ with canonical catalases to promote their activities and identify its co-factors, which could provide clues for precisely modifying TaWD40-4B.1 to fine-tune catalase activity and promote its application potential in molecular breeding.

Wheat is an allohexaploid, possessing A, B, and D subgenomes, formed via two inter-species cross and genome duplication of three diploid progenitors. Our Sanger sequencing indicates the alleles of TaWD40-4B.1 in the A and D subgenomes (TraesCS4A02G242800 and TraesCS4D02G071100) also have no truncated haplotype. The collinearity analysis shows that TaWD40-4B.1 appears to be the duplicated copy of TraesCS4B02G072300, and the duplication occurred during the formation of the tetraploid progenitor of wheat (Supplementary Fig. 18). TaWD40-4B.1 is identical to the homologs (TRIDC4BG011100 and TRITD4Bv1G024340) of tetraploid emmer and durum wheat. Previous studies found that both TRIDC4BG011100 and TRITD4Bv1G024340 have no truncated haplotype based on genome resequencing and whole-exome capture methods[70–73] (Supplementary Data 12). Oppositely, in hexaploid wheat, the TaWD40-4B.1$^T$ allele is present in approximately 20% of cultivars, but in 7.9% of Chinese and 15.4% of worldwide landraces (Fig. 8c and Supplementary Fig. 17). This suggests that the allelic variation may occur in hexaploid wheat, but it could not be excluded that the variation may occur in the tetraploid progenitor of wheat because there are fewer tetraploid wheat accessions in public genetic variation databases[70–73]. Consequently, the allelic variation may be possibly selected during the breeding of hexaploid wheat cultivars with high yield and suitable height, which is partially associated with that the drought-tolerant QTL qDSI4B.1 in chromosome 4BS is highly linked with the QTLs and genes (e.g., Rht1) governing grain yield and plant height (Supplementary Fig. 3), the major agronomic traits of wheat breeding. Consistent with this, the leading cultivars such as Jimai 22 and Yangmai 18 in China carry the TaWD40-4B.1$^T$ haplotype (Supplementary Data 8). On the other hand, the proportion of TaWD40-4B.1$^C$ is negatively correlated with the annual rainfall, and more accessions carrying TaWD40-4B.1$^C$ have been bred in the districts with less annual rainfall and abundant water resource (Fig. 8d, e). This indicates that the drought-sensitive haplotype appears to be less selected in the districts with more serious water deficiency, and TaWD40-4B.1$^C$ contributes to the drought tolerance of

wheat. Given the introgression of TaWD40-4B.1$^C$ enhances wheat drought tolerance and has no trade-offs effect (Fig. 8a, b), the TaWD40-4B.1$^T$ allele is a candidate target for both precise genomic editing and TaWD40-4B.1$^C$ replacement to breed new cultivars and improve leading cultivars with higher water-saving capacity and water use efficiency following the increasing deficiency of water resources. A recent study proved that a WD40 gene KRN2 was convergently selected to improve grain yield via allelic variation in the upstream sequence to reduce expression in maize and rice[44]. And as mentioned above, the syntenic drought tolerance QTL qDT4B.1 are involved in the maize and barley[13]. Thus, the members of the WD40 family may be selected for the improvement of drought and yield-related traits during crop domestication and breeding.

## Methods

### Phenotyping of drought tolerance for GWAS

The association-mapping panel composed of 198 wheat accessions was used for drought tolerance (Supplementary Data 1). The drought tolerance assay was conducted in a greenhouse under 16-h light/8-h dark and 16 °C/14 °C with a relative humidity of 40%. For each entry, 15 uniform seedlings were planted in a soil-filled plastic cone of 9.0 cm in diameter and 23.5 cm in depth. Each accession had five replicates. All of the accessions were randomly placed according to the completely randomized design. Water was withheld once the seedlings reached the three-leaf stage. The plants were re-watered until approximately 2/3 of entries were wilted when soil water content decreased to 15–20%. Two days after rewatering, a drought wilting score was assigned using a 1–6 scale illustrated in Fig S2, where score 1 represented a completely wilted plant, 2 a plant in which the first three leaves were fully wilted and the fourth was partially wilted, 3 a plant in which the first three leaves were fully wilted, 4 a plant in which the first two leaves were fully wilted and the third leaf was partly wilted, 5 a plant in which the first two leaves were fully wilted and the third not wilted, and 6 a plant in which only the second leaf was partly or hardly wilted. The index of drought tolerance was recorded according to the wilting score as illustrated in Supplementary Fig. 2. The DT index that was extremely deviated from those of the other replicates was considered as the outlier and excluded from the analysis.

### Genome-wide association study

All of the 198 accessions were profiled using the wheat 660 K genotyping array by Capital Bio Technology Co. Ltd. (Beijing, China). The polyploid version of the Affymetrix Genotyping Console software (Affymetrix, Santa Clara, CA) was used to conduct SNP allele clustering and genotype calling on the raw SNP data. The physical locations of SNPs were identified based on the IWGSC wheat genome sequence (IWGSC RefSeq v1.0). After filtering SNPs with a missing rate of more than 10% or with a minor allele frequency (MAF) of <5% using the PLINK software[74], the filtered SNPs were 419,606 retained for GWAS with the whole population[10]. Population structure was investigated using the ADMIXTURE software[75] and evaluating each K from 2 to 5. The stack graphs visualizing the population structure data were generated using the R script. The stack graphs visualizing the population structure data were generated using the R script. Principal components and kinship analysis (PCA) of the population were performed using the software GCTA[76]. Heat maps of kinship were generated based on the K-matrix using the pheatmap R package. The SNP genotypes, Q matrix, and trait scores for 198 accessions were incorporated into a compressed mixed linear model[77] implemented in the GAPIT R package[78]. The Phenotypic Variance Explanation (PVE) of each SNP and the kinship matrix used in this analysis were also automatically generated by GAPIT. Manhattan plots and quantile–quantile plot (Q–Q plot) were plotted by the qqman package in R script to present the results of association with the individual trait and important P value distributions, respectively. After a Bonferroni-adjusted correction, the P value for the suggestive

threshold was set to $2.3832 \times 10^{-6}$ (1/419,604). LD and haplotype blocks were constructed using the LDBlockShow software[79].

## Quantitative real-time PCR

The leaves and/or roots of young seedlings with and without treatment of water-withholding, $H_2O_2$, salt, alkali, and heat as well as the flag leaves, spikes, and grains of plants at the reproductive stage were sampled for RNA extraction. Total RNA was extracted using TRIzol reagent (Invitrogen, now ThermoFisher Scientific) following the supplier's protocol. Total RNA was treated with DNase I to remove DNA and then used to synthesize the first strand of cDNA using the FastKing RT Kit (Tiangen, https://en.tiangen.com). The single-strand cDNA was used as the template for quantitative real-time (qRT)-PCR based on TransStart® Green qPCR SuperMix (Transgen, AQ101). The gene encoding the β subunit of tubulin (*TaTub-β*, *TraesCS3D02G326900*) was used as the reference sequence. The relevant primer sequences are given in Supplementary Data 13.

## Vector construction and wheat transformation

The coding sequences of *TaWD40-4B.1^C* and *TaWD40-4B.1^T* were amplified with the cDNAs of cv. ShanRong 3 (SR3) and cv. YangMai 20 (YM20) as the template, respectively. To generate the overexpression lines, the coding sequences of *TaWD40-4B.1^C* and *TaWD40-4B.1^T* were ligated into the pGA3426 vector driven by the Ubiquitin promoter of *Zea mays*[80]. To generate the RNAi lines of *TaWD40-4B.1*, the sense and antisense fragment (532nd–1001st nt) of *TaWD40-4B.1* CDS was inserted into the pTCK303 vector driven by the Ubiquitin promoter of *Zea mays*. To generate the RNAi lines of canonical *TaCATs*, the sense and antisense fragments covering the conserved CDS regions of *TaCAT1s* (1070th–1350th nt) and *TaCAT3s* (430th–724th nt) in series were inserted into vector pTCK303. These constructs were transformed into *Agrobacterium tumefaciens* strain GV3101, and all of these transformants were transformed into both SR3 and YM20 with the *A. tumefaciens*–mediated shoot apical meristem transformation method[81]. The positively transformed lines were detected via genomic PCR with the total DNA extracted from the leaves as the template. The RNAi lines of *TaCATs* were crossed with *TaWD40-4B.1^C* and *TaWD40-4B.1^C* OE lines to generate *TaWD40-4B.1^C* OE/*TaCATs* RNAi lines and *TaWD40-4B.1^T* OE/*TaCATs* RNAi lines. The expression of *TaCAT1s*, *TaCAT3s*, *TaWD40-4B.1^C* and *TaWD40-4B.1^T* were measured with qRT-PCR as mentioned above.

## Water-withholding assay, $H_2O_2$, salt, alkali, heat, and hydroxyurea treatments

Equal weights of dry soils were placed in cuboid plastic pots of 12 cm in width and 10 cm in height. The soils were watered adequately, and fifty seeds were sawed in each pot. The pots were placed in the greenhouse under 16-h light (200 µmol m$^{-2}$ s$^{-1}$)/8-h dark and 22 °C/20 °C with a relative humidity of 40%. When wheat seedlings grew to approximately 5 cm in height, 25 uniformed seedlings were retained, and the other seedlings were removed. After growing in well-watered conditions for 2 weeks, the wheat seedlings were subject to water withholding for a further 14 days. After rewatering for 3 days, the seedlings were phenotyped, and the area and water content of wilted leaves as well as the shoot weight were measured.

The other treatments were conducted using the seedlings raised by hydroponics. Grains were imbibed on moist filter paper at 20 °C for 3 days, and the seedlings were removed to half-strength Hoagland's liquid medium (pH 6.0) in a greenhouse under 16-h light (200 µmol m$^{-2}$ s$^{-1}$)/8-h dark and 22 °C/20 °C with a relative humidity of 40%, and the medium was replaced every two days to keep fresh. The seedlings at the three- or one-leaf stage were used for phenotyping. For the treatments of $H_2O_2$, salt, and alkaline salt stress, wheat seedlings at three-leaf stage were transferred to the medium with or without 100 mM $H_2O_2$, 100 mM NaCl, and 50 mM NaHCO$_3$

(pH 10) for 7 days. For the treatment of heat stress, wheat seedlings at three-leaf stage were transferred into a growth chamber at 40 °C for 36 h followed by a 2-day recovery under the normal condition. For hydroxyurea treatment, wheat seedlings at one-leaf stage were transferred to medium with or without 1, 2, and 3 mM hydroxyurea (Beyotime, S1961) in a greenhouse under 16-h light (600 µmol m$^{-2}$ s$^{-1}$)/8-h dark and 22 °C/20 °C with a relative humidity of 40% for 5 days. After different periods (0–24 h) of treatment by $H_2O_2$, salt, alkaline salt, and heat stress, the leaves were sampled at the same time point for extracting RNA to avoid the influence of rhythm on the expression of genes. The seedlings raised in the medium were dehydrated by placing them in the air at room temperature for different periods (0–24 h), and the leaves and roots were sampled for extracting RNA at the same time point.

## Expression of TaWD40-4B.1^C/TaWD40-4B.1^T and TaCAT3A in yeast

The vector pESC-HIS (Agilent, USA) was used to express cMyc-tagged TaCAT3A and FLAG-tagged TaWD40-4B.1^C/TaWD40-4B.1^T in yeast (*Saccharomyces cerevisiae*) strain AH109 MATa (trp1-901, leu2-3, 112, ura3-52, his3-200) under the control of the GAL1 and GAL10 promoters, respectively. The coding sequence of the cMyc epitope EQK-LISEEDL was ligated into the multiple cloning sites (MCS) I downstream of the GAL1 promoter, and the open reading frame of *TaCAT3A* was inserted in front of the epitope sequence of cMyc. Similarly, the coding sequence of the FLAG epitope DYKDDDDK was ligated into the multiple cloning sites (MCS) II downstream of the GAL10 promoter, and the open reading frame of *TaWD40-4B.1^C/TaWD40-4B.1^T* was inserted in front of the epitope sequence of FLAG. The LiCl-PEG-based transformation and galactose-induced expression of the constructs ligated with cMyc-TaCAT3A alone, FLAG-TaWD40-4B.1^C/FLAG-TaWD40-4B.1^T alone, or cMyc-TaCAT3A and FLAG-TaWD40-4B.1^C/FLAG-TaWD40-4B.1^T were implemented according to the Instruction Manual of pESC Yeast Vectors. The Expression of TaCAT3A and TaWD40-4B.1^C/TaWD40-4B.1^T in yeast cells was detected via the Western-blot method with commercial c-Myc (ABclonal, AE010, 1:5000 (v/v)) and FLAG (ABclonal, AE005, 1:5000 (v/v)) antibodies.

## The measurement of catalase activity and growth rate of yeast cells

To determine the effect of TaWD40-4B.1^C and TaWD40-4B.1^T on the activity of TaCAT3A in yeast, the yeast lines expressing *cMyc-TaCAT3A* and/or *FLAG-TaDW40-4B.1^C/FLAG-TaWD40-4B.1^T* were harvested. The total proteins of yeast lines were extracted using the One Step Yeast Active Protein Extraction Kit (Sangon Biotech, C500026), and the concentration of proteins was calculated via the Bradford method. Then 1 mg of total protein was used for measuring catalase activity using Catalase (CAT) assay kit (Nanjing Jiancheng Bioengineering Institute, A007-1-1). For the growth rate measurement, the yeast lines expressing the above-mentioned pESC vectors were incubated in liquid YNB-His$^-$/Gal 2% medium at 30 °C till OD600 reached 0.1. Then $H_2O_2$ was added to the liquid medium to the final $H_2O_2$ concentration of 3 mM. The yeast lines were continued to incubate in the liquid medium with and without $H_2O_2$, and the OD600 values were recorded hourly. The curved lines based on OD600 values were drafted to mirror the growth rates.

## Yeast two-hybridization assay

Yeast two-hybridization assays were performed in strain AH109 using the LiCl-PEG method based on the vectors pGADT7 and pGBKT7 according to the manufacturer's protocol (Clontech). *TaCATs* was ligated into pGADT7, and *TaWD40-4B.1^C/TaWD40-4B.1^T* was ligated into pGBKT7. The pGADT7 and pGBKT7 constructs were co-transformed into AH109. Transformants were selected on the synthetic defined (SD) medium lacking Leu and Trp. Positive colonies were transferred to SD medium lacking Leu, Trp, and His.

## Split-luciferase complementation imaging

To compare the interaction of TaCATs and TaWD40-4B.1, the CDS of *TaCATs* were ligated into the vector JW771 to construct N-terminus luciferase (nLuc)-TaCATs fused ORF, the CDS of *TaWD40-4B.1*[C]/*TaWD40-4B.1*[T] were ligated into the vector JW772 to construct C-terminus luciferase (cLuc)-TaWD40-4B.1[C]/TaWD40-4B.1[T] ORF, and nLuc-GFP and cLuc-GFP were constructed as the negative control. To analyze the effect of TaWD40-4B.1 on the interaction between TaCATs, nLuc-TaCATs and cLuc-TaCATs were constructed, and CDS of *TaWD40-4B.1*[C]/*TaWD40-4B.1*[T] and *GFP* were ligated into the vector pRI101-AN. Corresponding constructs were transformed into *A. tumefaciens* strain EHA105 and the transformants were infiltrated into the leaves of *N. benthamiana* plants under the condition of 16-h light/8-h dark and 22 °C/20 °C with a relative humidity of 80% for three days. Then, the leaves were incubated with 1 mM Dual-luciferin-free acid (GoldBio, L-123) dissolved in 0.01% (v/v) Triton X-100 and incubated for 5 min in the dark. The fluorescence image was acquired with a CCD camera (Tanon-5200). In this work, to compare the difference in interaction capacity of TaWD40-4B.1[C]/TaCATs from TaWD40-4B.1[T]/TaCATs, the respective constructs with the same amount were combined in a 1:1 ratio.

## Co-immunoprecipitation assay

For the Co-IP assay, *A. tumefaciens* containing the various constructs were grown to an optical density (OD) of 0.8 and then infiltrated into the leaves of 4-week-old *N. benthamiana* plants. After 48–60 h of incubation, *N. benthamiana* leaves infiltrated with *A. tumefaciens* harboring either p35S::TaCAT-3A-GFP (driven by pRI101-GFPn) or p35S::TaWD40-4B.1[C/T]-cMyc (driven by pRI101-cMyc) were harvested, snap-frozen, and ground to a powder. Proteins were extracted from the powdered material in Cell lysis buffer for Western and IP (Beyotime, P0013). Following centrifugation (20,000 × g for 10 min at 4 °C), the supernatant was incubated at 4 °C for 4 h in the presence of monoclonal anti-MYC antibody-conjugated beads (MYC-Trap, Chromotek, ytma-100). The beads were rinsed six times in Cell lysis buffer for Western and IP (Beyotime, P0013). Proteins were eluted from the beads by boiling in SDS-PAGE Sample Loading Buffer (Beyotime, P0015A) for 10 min followed by western blotting using anti-MYC (Abclonal, AE010, 1:5000 (v/v)) or anti-GFP (Abclonal, AE012, 1:5000 (v/v)) antibody.

## Protein expression and purification

TaWD40-4B.1[C] and TaWD40-4B.1[T] were cloned into pET28a to generate TaWD40-4B.1[C]-His and TaWD40-4B.1[T]-His, while TaCAT3A was cloned into pMAL-C2X to get TaCAT3A-MBP. *Escherichia coli* strain DE3 was transformed using recombinant plasmids and the vector separately, and cultured at 37 °C in 500 mL LB. When the OD600 reached 0.6, protein expression was induced at 16 °C with 0.8 mM isopropyl β-D-1-thiogalactopyranoside (Beyotime, ST098) for 16 h. Purification of the recombinant proteins was performed following the manufacturer's instructions.

## Pull-down assay and the measurement of in vitro catalase activity

Two mg MBP or four mg MBP-tagged TaCAT3A proteins were immobilized on Amylose resins. Immobilized beads were incubated with 1 mg His-TaWD40-4B.1[C/T] in 50 mM KH₂PO₄ (pH 7.4) for 1 h at 4 °C. After centrifugation at 60 × g at 4 °C for 1 min, the supernatant was removed, and the beads were washed six times with pre-cooled 50 mM KH₂PO₄ (pH 7.4). The resin-retained proteins were analyzed by immunoblotting using anti-His (Abclonal, AE003, 1:5000 (v/v)) or anti-MBP antibodies (Abclonal, AE016, 1:5000 (v/v)). To assess the effect of TaWD40-4B.1[C] and TaWD40-4B.1[T] on TaCAT3A activity, various concentrations of purified TaWD40-4B.1[C] or TaWD40-4B.1[T] proteins were mixed with TaCAT3A in 50 mM KH₂PO₄ (pH 7.4) for 1 h at 4 °C, and

catalase activity was then determined using Catalase (CAT) assay kit (Nanjing Jiancheng Bioengineering Institute, A007-1-1).

## TaCAT oligomerization assay based on cross-linked proteins

The SDS-PAGE electrophoresis using cross-linked proteins was implemented to reveal the attendance of oligomerized TaCATs in vivo and in vitro according to the previous report[82] with slight modifications. Specifically, proteins were extracted from wheat seedlings at 4 °C in 50 mM KH₂PO₄ (pH 7.4) containing 1 mM PMSF (Beyotime, ST505) and 1× Protease inhibitor cocktail (Beyotime, P1005). After elimination of cell debris by centrifugation (12,000 × g) at 4 °C for 10 min, 2 mM dithiobis (succinimidyl propionate) (DSP; Sangon Biotech, C110213) was added to the extract, which was kept at 4 °C with shaking for 30 min. As for the analysis of in vitro oligomerization, the purified TaCAT3A-MBP by prokaryotic expression was mixed with various concentrations of purified TaWD40-4B.1[C] or TaWD40-4B.1[T] proteins in 50 mM KH₂PO₄ (pH 7.4) for 1 h at 4 °C, and DSP was added to cross-link the oligomeric proteins in the same way. The crosslinker was then quenched with 50 mM Tris buffer (pH 7.6) for 15 min at room temperature. Protein sample buffer containing 5% β-ME (reversal of crosslink) or without reducing agent (no reversal of crosslink) was added, and samples were boiled for 10 min before loading.

## Subcellular localization and bimolecular fluorescence complementation (BiFC)

For subcellular localization, pBI221-GFP-TaCATs, pBI221-GFP-TaWD40-4B.1[C] and pBI221-GFP-TaWD40-4B.1[T] constructs were built, and each of them along with the plastid harboring peroxisome marker DsRed-SKL were introduced into wheat protoplasts by the PEG-mediated transfection[83]. For BiFC, the coding sequence of TaWD40-4B.1[C/T] and TaCATs was introduced into the pUC-SPYCE(MR) and pUC-SPYNE(R) 173 vector, respectively[84], and GUS-YFPC was constructed as the negative control. Corresponding constructs along with the plastid harboring peroxisome marker DsRed-SKL were co-transformed into wheat protoplasts using the same method. Then the protoplasts were incubated in dark at 25 °C for 16–24 h. The confocal images were finally captured using the ZEISS LSM 900 system.

## Isolation of intact peroxisomes

The protoplasts co-expressing GFP-tagged proteins and RFP-SKL mentioned above were harvested and peroxisomes were isolated as described[85] with some modifications. Briefly, 5 × 10⁶ intact peroxisomes were centrifuged at 5,000 × g for 1 min at 4 °C in grinding buffer (170 mM Tricine-KOH, 1 M sucrose, 1% [w/v] BSA, 2 mM EDTA, 5 mM DTT, 10 mM KCl, 1 mM MgCl₂, and 1× protease inhibitor cocktail, pH 7.5) to obtain the crude extract. The supernatant was loaded onto Percoll density gradients prepared in TE buffer (20 mM Tricine-KOH and 1 mM EDTA, pH 7.5)[85], and centrifuged for 12 min at 13,000 × g followed by centrifugation for 20 min at 27,000 × g. After centrifugation, peroxisomes are located at the bottom of the gradients. Fractions were collected from the top and bottom of the centrifugation tubes and washed in 36% (w/w) sucrose in TE buffer and centrifuged for 30 min at 39,000 × g. The top fraction was labeled as the cytosol and the pellet as the crude peroxisome. Subsequently, the crude peroxisome was loaded onto a sucrose density gradient (2 mL 41% [w/w], 2 mL 44% [w/w], 2 mL 46% [w/w], 3 mL 49% [w/w], 1 mL 51% [w/w], 1.5 mL 55% [w/w], and 1 mL 60% [w/w] in TE buffer) and centrifuged at 80,000 × g for 2 h. After centrifugation, a white band appeared at the interface of 55 and 51% sucrose. The white band was collected and labeled peroxisome extract. These fractions were resuspended and boiled in the same volume of 1× SDS-PAGE Sample Loading Buffer (Beyotime, P0015A) and subjected to immunoblot analysis using anti-GFP (Abclonal, AE012, 1:5000 (v/v)) and anti-RFP (Abclonal, AE020, 1:5000 (v/v)) antibodies.

## Analysis of MDA and $H_2O_2$ content, and catalase activity of wheat seedlings

Two-week-old seedlings grown in the soil were subject to water with-holding for 1 week. The second leaves of water-withheld and well-watered seedlings were sampled for the following analysis. The relative quantification of $H_2O_2$ was manifested based on 3,3-diaminobenzidine (DAB) staining. The leaves were stained in $1\,mg\,mL^{-1}$ DAB (Coolaber, CD4181) solution in the dark for 12 h. After staining, the leaves were bleached with 85% ethanol in a water bath at 100 °C until chlorophyll completely faded. The leaves were snap-frozen in liquid nitrogen for the measurement of MDA content and in vivo catalase activity using the Lipid Peroxidation MDA Assay Kit (Beyotime, S0131) and Catalase (CAT) assay kit (Nanjing Jiancheng Bioengineering Institute, A007-1-1) according to the suppliers' protocols.

## The yield trail and the measurement of yield-associated traits

Germinated seeds of transgenic lines and non-transgenic controls by placing them at room temperature for 3 days were transferred into plastic pots ($10 \times 10 \times 14$ cm) containing an equal weight of soil. One seed was plated in one pot, and each line had twenty pots. Each pot was watered with the same amount of water to maintain the uniformity of soil-relative water content every 2 days until the beginning of jointing. Then, the seedlings were randomly separated into two groups for the parallel water-limited and well-watered treatment. The water-limited treatment was conducted via watering every ten days, while the well-watered treatment was performed by watering every two days. The watering was terminated at the beginning of senesced yellowing. After harvesting, the yield as well as the yield-associated indices including grain size and width, 1000-grain weight, grain number per spike/plant, and plant height were measured. During grain filling, $CO_2$ assimilation rate and transpiration rate were measured with an LI-6400XT Portable Photosynthesis System (Li-Cor, USA). The water use efficiency was calculated by dividing the $CO_2$ assimilation rate by the transpiration rate. The grain yield and yield-associated indices were recorded based on 10 seedlings per line, and the $CO_2$ assimilation rate and transpiration rate had approximately 20 replicates. The field water-saving trial was conducted at Shijiazhuang, China (E114°32′18″, N38°7′10″) from Oct, 2021 to Jun, 2022. In each irrigation field, all lines were planted randomly in blocks of 1.5 m × 9 m, and each line was planted in three blocks. Two irrigation treatments were conducted, one was that no water was irrigated during the whole life course, and the other was that $750\,m^3\,ha^{-1}$ water was irrigated at the sowing and reviving stages respectively (in total $1500\,m^3\,ha^{-1}$). The spike density was measured at the filling stage, and the grain yield and other indices were measured after harvesting.

## Construction of near-isogenic lines

Drought-tolerant cultivar Luyuan 301 carrying the $TaWD40\text{-}4B.1^C$ allele and drought-sensitive cultivar Chuanmai 36 carrying the $TaWD40\text{-}4B.1^T$ allele were crossed each other to obtain F1 seeds. Then Luyuan 301 and Chuanmai 36 were used as recurrent parents to construct their near-isogenic lines (NILs) carrying $TaWD40\text{-}4B.1^T$ and $TaWD40\text{-}4B.1^C$ alleles, respectively. $TaWD40\text{-}4B.1$ was genotyped in each successive generation, and the heterozygous hybrids were backcrossed to create the BC5F1 population. Premature egg cells of BC5F1 populations were fertilized with maize pollen to construct haploid seedlings, which were used to construct double haploids by chromosome doubling. The near-isogenic lines of Luyuan 301 and Chuanmai 36 carrying $TaWD40\text{-}4B.1^T$ and $TaWD40\text{-}4B.1^C$ alleles, respectively, were genotyped from double haploids. The drought tolerances of NILs were detected by the water withholding assay as mentioned above.

## Geographical distribution of wheat varieties

For the geographical distribution of wheat varieties, 1912 varieties were projected on the map according to their origin information[55–58,86], among which 600 were from China. The haplotypes of 600 varieties from China were confirmed via Sanger sequencing. The haplotypes of the other 1312 varieties were referred to the previous studies[55–58] that are integrated in SnpHub[87], and the haplotypes of 57 were confirmed via Sanger sequencing. The average annual rainfall (1973–2019) of provinces/autonomous regions of China were calculated according to the data provided by the National Centers for Environmental Information (https://www.ncei.noaa.gov/). Pearson correlation analysis was used to evaluate the association between $TaWD40\text{-}4B.1^T$ allele frequency with average annual rainfall based on 583 wheat varieties from 16 provinces/cities of China.

## Domain and 3D structure prediction and the phylogenetic analysis

The domain of $TaWD40\text{-}4B.1^C$/$TaWD40\text{-}4B.1^T$ and the paralogs were analyzed with SMART (https://smart.embl-heidelberg.de/). Three-dimensional structures of $TaWD40\text{-}4B.1^C$ and $TaWD40\text{-}4B.1^T$ were predicted with SWISS-MODEL (https://swissmodel.expasy.org/). The peptide sequences of TaCATs were aligned in ClusterX2, and the phylogenetic tree was drafted with MEGA7.0.

## Statistics and reproducibility

The normal distribution of the data was measured with the Shapiro–Wilk test. The difference among the data coinciding with normal distribution was conducted with a two-sided z-test, two-sample dependent two-sided t test, two-sample independent two-sided t test, or one-way ANOVA test. The post hoc comparison of one-way ANOVA was conducted with the Tukey method at a significance level of 0.05. The correlation between the proportion of $TaWD40\text{-}4B.1^T$ and annual rainfall was conducted with the Pearson analysis. All experiments were repeated independently three times.

## Reporting summary

Further information on research design is available in the Nature Portfolio Reporting Summary linked to this article.

## Data availability

Data supporting the findings of this work are available within the paper and its Supplementary Information files. A reporting summary for this article is available as a Supplementary Information file. Source data are provided with this paper.

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

## Acknowledgements

We are grateful to Dr. Changle Ma from Shandong Normal University for presenting the plastid of peroxisome marker DsRed-SKL and Dr. Tongjin Zhao from Fudan University for the critical advice on this work. The work was supported by the National Key Research and Development Program (2022YFF1001600 to M.W.), the Key Project of Natural Science Foundation of Shandong (ZR202105200003 to M.W.), the National Natural Science Foundation of China (31870242 and 32170297 to M.W. and 31720103910 to G.X.), and the National Transgene Project (2020ZX08009-11B to M.W.).

## Author contributions

G.X. and M.W. conceived the work. G.T., S.W., M.W., Y.W., X.W., J.W., S.L., and D.H. conducted the experiment and analyzed the data. M.W., G.X., and G.T. wrote the paper. All authors read the manuscript.

## Competing interests

The authors declare no competing interests.
