## [Peer Review File · Nature Communications]

Allelic variation at the TaWD40-4B.1 locus contributes to drought tolerance by modulating catalase activity in wheatReviewers' Comments:

Reviewer #1:

Remarks to the Author:

This manuscript begins by describing a mutant isolated by GWAS analysis, and authors clone an important drought-tolerant gene, TaWD40-4B.1. Further analyses suggest that TaWD40-4B.1 has been selected in the breeding process in response to drought stress. The TaWD40-4B.1 protein interacts with catalase. The authors show that the mutant has low catalase activity. The authors then conclude that "TaWD40-4B.1 contributes to drought tolerance via directly modulating catalase activity in wheat". However, the interpretation is overly categorical and incomplete. Does TaWD40-4B.1 indeed regulate catalase activity and/or other responses as a result of drought stress? Is the drought stress response phenotype observed in TaWD40-4B.1 mutant related to the loss of catalase activity, and/or the result of additional defects unrelated to the interaction of WD40-4B.1 and catalase?

Major points:

1. the cellular localization of TaWD40-B, the place of the interaction between TaWD40 and CAT3A require more detailed cell biology and biochemistry evidences.
2. more biochemisry studies are required to demonstrate how catalase activity is enhanced by the interaction between TaWD40 and CAT3A.
3. if the phenotype of TaWD40-B mutant is due to lower catalase activity, other phenotypes, such as salt, cold etc, should be tested.
4. The catalase mutant phenotype is absent or much alleviated in plants grown at low light or at high CO₂ but under standard growth conditions. It would be useful to know what the phenotype of TaWD40-B mutant is under hydroxyurea (HU) treatment and whether it depends on photorespiration (low light/high CO₂ conditions). If so, this would strengthen the genetic link between TaWD40-B function and catalase.
5. no genetic evidence to show TaWD40-B mediated drought tolerance links to catalases

minor points:

- 1, few controls are missing.
- 2, nLuc, cLuc should not be used as a control.

Reviewer #2:

Remarks to the Author:

Drought stress severely affects crop growth, restricting crop production. Identification of genes that can promote crop drought stress tolerance but not affect crop growth is of great significance for agricultural production. In this study, Tian et al identified an drought tolerant allele TaWD40-4B.1 located on the quantitative trait locus qDSI.4B.1 using the GWAS. Interestingly, they found that a nonsense variation of the gene led to an allele encoding a truncated peptide TaWD40-4B.1T, and that full-length allele TaWD40-4B.1C but not TaWD40-4B.1T confers wheat drought tolerance as TaWD40-4B.1C but not TaWD40-4B.1T interacts with catalase and repressing ROS overaccumulation in wheat under drought. Moreover, they proposed a negative correlative between the annual rainfall in different areas and the proportion of TaWD40-4B.1C allele in wheat accessions, which may be useful for the selected breeding of drought-tolerant wheat in arid regions.

Overall, the study presented an interesting story with novel findings advancing our understanding of the role of TaWD40-4B.1 in wheat drought stress tolerance. I have several major and minor concerns about this manuscript.

1. Authors showed that TaWD40-4B.1C has physical interaction with canonical catalases and enhances

their H₂O₂-degradating activity, and wheat plants carrying TaWD40-4B.1C but not TaWD40-4B.1T have higher catalase activity and lower H₂O₂ contents. However, it is still unclear whether TaWD40-4B.1C-conferred drought stress tolerance is really caused by the activation of catalase, as only H₂O₂ content and catalase activity but no genetic experiments or pharmaceutical rescue treatment were performed to get the relationships between TaWD40-4B.1 and catalase. Also, there is also no evidence to support the the correlative between catalase activity and variations of TaWD40-4B.1C and TaWD40-4B.1T in the tested wheat natural accessions.

2.I noticed that OE-T but not OE-C is tolerant to high H₂O₂ stress treatment, but authors did not test or discuss the possible involvement of TaWD40-4B.1C in other abiotic stresses, such as high salinity. Is TaWD40-4B.1 specific to drought stress? Is its expression induced by H₂O₂ and other abiotic stresses? Authors should test and discuss this.

3.Authors focused on the negative effects of H₂O₂ on plant drought stress tolerance. From another point of view, however, H₂O₂ is also necessary for the plant response to drought stress. Recently, it is reported that H₂O₂ promotes drought stress tolerance by increasing ABA accumulation through sulfenylating a tryptophan synthase b subunit, and thus may be involved in the coordination of plant growth and stress response (Liu et al., 2022 Mol Plant 15: 1-18). Therefore, the regulation of catalase activities and multiple roles of H₂O₂ in plant drought stress response and tolerance should be described in the Introduction and discussed in the Discussion.

4.Figures S10 and 3 show that TaWD40-4B.1 (C or T) is localized in cytosol while TaCAT in peroxisomes. However, the interaction of TaWD40-4B.1 with TaCAT occurs both in the cytosol and peroxisomes. How to explain such results? In addition, the Figure 3B (BiFC) cannot rule out the possibility that the small puncta is other organelles or structures other than peroxisomes as no peroxisomal marker was used.

5.Does TaWD40-4B.1 affect TaCATs expression?

6.The second leaves of the treated and untreated wheat seedlings were employed to analyze the H₂O₂ content, CAT activity and MDA content. What's the expression pattern of TaWD40-4B.1 in different tissues and organs of wheat?

7.Does TaWD40-4B.1 has paralogs in A- or D-subgenome in wheat? If yes, are there natural variations in their gene structures, similar to the truncation found in TaWD40-4B.1C but not TaWD40-4B.1T?

8.Line 256-265 "Given that" was used two times in one paragraph. Please consider to change one of them as "Considering that" or "Because" etc.

9.Line 275 and 277 "BMP-TaCAT3A" should be "MBP-TaCAT3A"?

10.Authors should provide the category numbers for the antibodies, kit and and important reagents.

11.Figure 4C the "H₂O₂" in the y axis should be lowercase.

12.Figure S12A, "Well withheld" is wrong.

13.Figure S12K, the "mmol H₂O₂" in the y-axis is wrong. Please carefully check all the labeling in the figures.

Reviewer #3:

Remarks to the Author:

This manuscript describes an interesting characterisation of the role of TaWD40-4B.1 in drought tolerance in wheat. GWAS analysis revealed that drought tolerance was associated with the qDT4B locus on chromosome 4BS. Drought tolerance is associated with the WD40 gene TaWD40-4B.1. Wheat accessions carrying the truncated allele TaWD40-4B.1T have a lower drought tolerance than those with complete allele TaWD40-4B.1C. Modifications in TaWD40-4B.1C expression had no effect on plant growth and grain yield under water-replete conditions. Data are presented showing that the allelic nonsense variant of TaWD40-4B.1, produces a truncated peptide TaWD40-4B.1T, which is associated with the lower drought tolerance in wheat accessions. In contrast, the complete allele TaWD40-4B.1C is associated with enhanced drought tolerance and higher grain yields under drought stress. Crucially, TaWD40-4B.1C (but not TaWD40-4B.1T) interacts with the catalase proteins enhancing in vivo catalase activities and the tolerance to oxidative stress under drought. WD40 proteins function as scaffolds for protein-protein interactions. In the present study, TaWD40-4B.1C is suggested to bind catalase in a manner that enhances drought tolerance by increasing the catalytic activity of the enzyme. However, this study provides little indication of the mechanisms whereby the protein encoded by TaWD40-4B.1C can interact with this peroxisomal iron-containing homotetrameric protein to increase the dismutation of H₂O₂ into H₂O and O₂. A mechanistic study/explanation of where the TaWD40-4B.1C protein/catalase interaction takes place and how this interaction modifies catalase activity is essential in order to advance current knowledge. The catalase proteins in plants are known to undergo a number of post-translational modifications and numerous catalase binding partners have been reported in the literature. The present manuscript ignores the extensive literature on catalase binding proteins that modify biotic and abiotic stress tolerance traits.

I am concerned about the physiological relevance of some of the studies. Wheat seedlings were grown under well-watered conditions for two weeks and then subjected to drought by withholding water for 14 days. This is a very severe stress that will take the plants almost to death. At this point, the seedlings were re-watered and various shoot parameters were measured after 3 days. This study does not therefore examine the role of TaWD40-4B.1C in mild drought conditions but rather the ability to withstand drought-induced death.

Crucially also, the treatment with hydrogen peroxide involved culturing seedlings at the three-leaf stage with 100mM hydrogen peroxide for one week. This is an excessive amount of oxidant that would most certainly kill the seedlings.

The levels of hydrogen peroxide shown in Figure 5C are given as micromoles per gm fresh weight. The values are far too high. Values should be in nanomoles per gm fresh weight. I am therefore very concerned about the methods used for the extraction and assay of hydrogen peroxide, which is notoriously difficult to extract and assay without artefact (see for example: Noctor et al. (2016) Oxidative stress and antioxidative systems: recipes for successful data collection and interpretation. Plant Cell Environment. 39. 1140-1160. Since the effects of TaWD40-4B.1C on the ability of catalase to prevent accumulation of hydrogen peroxide, accurate data on this metabolite are essential.

The authors state that the relative quantification of hydrogen peroxide was based on 3,3-diaminobenzidine (DAB) staining but this method is qualitative and not quantitative. To make this assay quantitative, internal standards of known hydrogen peroxide concentration are required, and this is not possible.

Reviewer #4:

Remarks to the Author:

Dear Authors,

Your study to investigate the influence of an allelic variation inside the TaWD40-1 4B.1 and how it can contribute to drought tolerance is a very impressive and comprehensive study.

I have a few comments to point out

why did you use 198 wheat accessions only?

have they geographically distributed over the world?

did you select them based on different biological status, breeding line, cultivars, landraces and wild?

how many SNP markers did you use in GWAS at the end?

it was not clear, how the GWAS strategy had been done, please find a reference for GWAS in cereals.

it was not clear how did you reach the specific allele.

did you resequence the gene in the accessions?

how can you elucidate your conclusion that TaWD40-4B.1 was selected by breeding?

Please make sure to provide proof of that assumption.

can you show the sequence differences between Yangmai 20 (YM20) carrying the 189 TaWD40-4B.1T allele and Shanrong 3 (SR3) carrying the TaWD40-4B.1C.

can you present the gene construct of RNAi.

can you provide the ANOVA or any other statistical analysis to confirm the results of grain yield-related traits under both drought and control conditions?

Response to the comments of four reviewers

Reviewer #1 (Remarks to the Author):

This manuscript begins by describing a mutant isolated by GWAS analysis, and authors clone an important drought-tolerant gene, TaWD40-4B.1. Further analyses suggest that TaWD40-4B.1 has been selected in the breeding process in response to drought stress. The TaWD40-4B.1 protein interacts with catalase. The authors show that the mutant has low catalase activity. The authors then conclude that “TaWD40-4B.1 contributes to drought tolerance via directly modulating catalase activity in wheat”.

However, the interpretation is overly categorical and incomplete. Does TaWD40-4B.1 indeed regulate catalase activity and/or other responses as a result of drought stress? Is the drought stress response phenotype observed in TaWD40-4B.1 mutant related to the loss of catalase activity, and/or the result of additional defects unrelated to the interaction of WD40-4B.1 and catalase?

Answer: Thank you for your advice. In our original submission, we found TaWD40-4B.1^C but not TaWD40-4B.1^T promoted *in vivo* catalase activity in wheat seedlings under drought (Figs 4, S12A). Especially, TaWD40-4B.1^C but not TaWD40-4B.1^T obviously promoted the *in vitro* catalase activity (Fig 4E), providing direct evidence for that TaWD40-4B.1^C can regulate catalase activity. In the revised submission, to confirm this relationship, we firstly constructed the knock-down lines of *TaCATs* with the RNAi method, and found that *TaCAT* knock-down reduced drought tolerance of wheat (Fig 6A). Then, we conducted the genetic analysis by crossing *TaCAT* RNAi lines and TaWD40-4B.1^C / TaWD40-4B.1^T OE lines, and found that *TaCAT* knock-down almost abolished the role of TaWD40-4B.1^C in drought tolerance, catalase activity and H₂O₂ level (Fig 6B-F). TaWD40-4B.1^C promoted the oligomerization of catalase (Fig 4F-I), and oligomerization has proved to be the fundamental mechanism of catalase maturation to enhance catalase activity. In line with these data, it could be concluded that TaWD40-4B.1^C enhances drought tolerance via, at least in large part, promoting catalase activity. This result has been added in the revised manuscript (Figs 6, S14; Lines 416-452), and the methods have been added in M&M (Lines 746-757 and 898-914).

Major points:

1, the cellular localization of TaWD40-B, the place of the interaction between TaWD40 and CAT3A require more detailed cell biology and biochemistry evidences.

Answer: Thank you for your advice. The cellular localization of *TaCATs* and TaWD40-4B.1 as well as the place of the interaction between *TaCATs* and TaWD40-4B.1 has been re-analyzed by co-expressing peroxisome marker DsRed-SKL in wheat protoplasts (Figs 3B, S9A, S10). Moreover, we have confirmed the subcellular localization of *TaCATs* and TaWD40-4B.1 with the western blotting

assay using the total, peroxisomal and cytoplasmatic proteins extracted from the protoplast (Fig S10B). The results have been added in the revised manuscript (Lines 271-284), and the methods had been updated and added in M&M (Lines 916-952).

2, more biochemistry studies are required to demonstrate how catalase activity is enhanced by the interaction between TaWD40 and CAT3A.

Answer: Thank you for your advice. The mechanism modulating catalase activity has been widely investigated in animal cells, the catalase activity can be affected by posttranslational modification such as phosphorylation and ubiquitination, binding of heme, and formation of oligomers. The formation of oligomers has proved to be the most common mechanism for activating catalase activity; oligomeric (majorly tetrameric) catalase has high activity, but the monomeric form has lower activity. In plant cells, a set of proteins interacting with catalase have been identified, but only several proteins were studied regarding the mechanisms for modulating catalase activity: phosphorylation (OsSTRK1, AtCPK8), ubiquitination (OsAPIP6), maintaining optical conformation by chaperone AtNCA1, promoting abundance by OsSRL10. We found TaWD40-4B.1^C promoted catalase activity in the *in vitro* enzymatic assay system that did not contain heme and lacked the condition for protein modification, which suggests that TaWD40-4B.1 may promote catalase activity via influencing oligomer formation but not posttranslational modification of catalase or affecting heme binding. Indeed, we found that TaWD40-4B.1^C but not TaWD40-4B.1^T promoted the homo-interaction between TaCAT and TaCAT, and the formation of oligomeric catalase holoenzymes *in vivo* and *in vitro* (Fig 4F-I), which is consistent with the previous study that rice catalase can form dodecameric holoenzyme containing 12 units with high activity and fast turnover rate of H₂O₂ (Zhang, et al. Molecular Plant, 2016). Thus, TaWD40-4B.1^C promoted the formation of oligomeric (dodecameric) catalase holoenzymes in wheat under both well-watered and drought conditions, suggesting the enhancement of TaWD40-4B.1 to catalase activity under drought condition is in relation to the oligomer (dodecamer) formation. The results and discussion have been added in the revised manuscript (Lines 351-377 in Results and lines 593-610 in Discussion).

3, if the phenotype of TaWD40-B mutant is due to lower catalase activity, other phenotypes, such as salt, cold etc, should be tested.

Answer: Thank you for your advice. We have conducted the phenotypes under salt, alkali and heat stresses, and found that TaWD40-4B.1^C but not TaWD40-4B.1^T also enhances the tolerance to salt, alkali and heat stresses (Fig S13). This implies that TaWD40-4B.1^C is an excellent haplotype for germplasm improvement with broad-spectrum tolerance to diverse abiotic stresses. The results and discussion have been added in the revised manuscript (Lines 410-413 in Results and lines 558-561 in Discussion), and the treatment method has been presented in M&M (Lines 771-781).

4, The catalase mutant phenotype is absent or much alleviated in plants grown at low light or at high CO₂ but under standard growth conditions. It would be useful to know

what the phenotype of TaWD40-B mutant is under hydroxyurea (HU) treatment and whether it depends on photorespiration (low light/high CO₂ conditions). If so, this would strengthen the genetic link between TaWD40-B function and catalase.

Answer: Thank you for your advice. High light intensity induces photorespiration, which leads to generation of large amounts of ROS (H₂O₂) in the peroxisomes, resulting in severe plant growth retardation. According to your valuable advice, we have conducted the phenotype analysis of hydroxyurea (HU) treatment using SR3 and the *TaWD40-4B.1^C* RNAi line under high light intensity (Fig S15). Under high light intensity condition, SR3 grew better than the RNAi lines in the absence of HU, whilst the growth difference became smaller with the increase of concentrations of HU treatment. This indicates that TaWD40-4B.1 can indeed modulate catalase activity. The results have been added in the revised manuscript (Lines 439-446), and the treatment method has been presented in M&M (Lines 781-784).

5, no genetic evidence to show TaWD40-B mediated drought tolerance links to catalases.

Answer: Thank you for your advice. We have conducted the genetic analysis by constructing the cross line of *TaCAT* RNAi lines and *TaWD40-4B.1^C* / *TaWD40-4B.1^T* OE line. The result indicates that *TaCAT* RNAi almost abolished the role of *TaWD40-4B.1^C* in drought tolerance (Fig 6). Please see the answer above for details.

minor points:

1, few controls are missing.

Answer: Thank you for your advice. The controls have been added in the BiFC assay for the interaction between TaCAT1A/2A and TaWD40-4B.1^{T/C} (Fig S9).

2, nLuc, cLuc should not be used as a control.

Answer: Thank you for your advice. We replaced nLuc and cLuc with nLuc-GFP and cLuc-GFP in the split-luciferase complementation imaging (SLCI) assay (Fig 3D, 4F, 4G). Besides, we replaced cYFP with cYFP-GUS in the bimolecular fluorescence complementation (BiFC) assay (Fig 3B, S9). The method of vector construction has been also updated (Lines 839-846, 922-926).

Reviewer #2 (Remarks to the Author):

Drought stress severely affects crop growth, restricting crop production. Identification of genes that can promote crop drought stress tolerance but not affect crop growth is of great significance for agricultural production. In this study, Tian et al identified an drought tolerant allele TaWD40-4B.1 located on the quantitative trait locus qDSI.4B.1 using the GWAS. Interestingly, they found that a nonsense variation of the gene led to an allele encoding a truncated peptide TaWD40-4B.1T, and that full-length allele TaWD40-4B.1C but not TaWD40-4B.1T confers wheat drought tolerance as TaWD40-4B.1C but not TaWD40-4B.1T interacts with catalase and repressing ROS

overaccumulation in wheat under drought. Moreover, they proposed a negative correlative between the annual rainfall in different areas and the proportion of TaWD40-4B.1C allele in wheat accessions, which may be useful for the selected breeding of drought-tolerant wheat in arid regions.

Overall, the study presented an interesting story with novel findings advancing our understanding of the role of TaWD40-4B.1 in wheat drought stress tolerance. I have several major and minor concerns about this manuscript.

1. Authors showed that TaWD40-4B.1C has physical interaction with canonical catalases and enhances their H₂O₂-degradating activity, and wheat plants carrying TaWD40-4B.1C but not TaWD40-4B.1T have higher catalase activity and lower H₂O₂ contents. However, it is still unclear whether TaWD40-4B.1C-conferred drought stress tolerance is really caused by the activation of catalase, as only H₂O₂ content and catalase activity but no genetic experiments or pharmaceutical rescue treatment were performed to get the relationships between TaWD40-4B.1 and catalase. Also, there is also no evidence to support the correlative between catalase activity and variations of TaWD40-4B.1C and TaWD40-4B.1T in the tested wheat natural accessions.

Answer: Thank you for your advice. To confirm this relationship, we constructed the knock-down lines of *TaCATs* with the RNAi method, and found that *TaCATs*' knock-down reduced drought tolerance of wheat (Fig 6A). Then, we conducted the genetic analysis by crossing *TaCAT* RNAi lines and TaWD40-4B.1^C / TaWD40-4B.1^T OE lines, and found that *TaCATs*' knock-down almost abolished the role of TaWD40-4B.1^C in drought tolerance, catalase activity and H₂O₂ level (Fig 6B-F). These results indicate that the drought stress response phenotype observed in TaWD40-4B.1 transgenic lines is, at least in large part, related to the change of catalase activity. Besides, hydroxyurea (HU) has proved to inhibit catalase activity. We found that under high light intensity condition, SR3 grew better than the *TaWD40-4B.1^C* RNAi line in the absence of HU, whilst the growth difference became smaller with the concentrations of HU treatment (Fig S15). This result indicates that TaWD40-4B.1 can indeed modulate catalase activity. The results have been added in the revised manuscript (Figs 6, S14, S15; Lines 416-452), and the methods have been added in M&M (Lines 746-757 and 781-784).

We have measured the catalase activities of 24 wheat accessions harboring *TaWD40-4B.1^C* and 24 wheat accessions harboring *TaWD40-4B.1^T* under drought (Table S5), and found that the catalase activities of the accessions harboring *TaWD40-4B.1^C* are higher than those of the accessions harboring *TaWD40-4B.1^T* under drought (Fig 6G). The results have been added in the revised manuscript (Lines 446-450).

2. I noticed that OE-T but not OE-C is tolerant to high H₂O₂ stress treatment, but authors did not test or discuss the possible involvement of TaWD40-4B.1C in other abiotic stresses, such as high salinity. Is TaWD40-4B.1 specific to drought stress? Is

its expression induced by H₂O₂ and other abiotic stresses? Authors should test and discuss this.

Answer: Thank you for your advice. *TaWD40-4B.1* was also induced by H₂O₂, salt, alkali and heat stresses (Fig S13C). We have conducted the phenotypes under salt, alkali and heat stresses, and found that *TaWD40-4B.1^C* but not *TaWD40-4B.1^T* also enhanced the tolerance to salt, alkali and heat stresses (Fig S13D-F). This implies that *TaWD40-4B.1^C* is an excellent haplotype for germplasm improvement with broad-spectrum tolerance to diverse abiotic stresses. The results have been added and discussed in the revised manuscript (Lines 403-413 and 558-560), and the treatment method has been presented in M&M (Lines 771-789).

3. Authors focused on the negative effects of H₂O₂ on plant drought stress tolerance. From another point of view, however, H₂O₂ is also necessary for the plant response to drought stress.

Recently, it is reported that H₂O₂ promotes drought stress tolerance by increasing ABA accumulation through sulfenylating a tryptophan synthase b subunit, and thus may be involved in the coordination of plant growth and stress response (Liu et al., 2022 Mol Plant 15: 1-18). Therefore, the regulation of catalase activities and multiple roles of H₂O₂ in plant drought stress response and tolerance should be described in the Introduction and discussed in the Discussion.

Answer: Thank you for your advice. H₂O₂ plays complicated roles in drought tolerance and other processes. As an important signaling molecule, H₂O₂ can trigger the signaling transduction network to produce physiological responses (for example stomatal closure) upon drought stress and is necessary for growth and reproduction. Besides, H₂O₂ also participates in the modification of proteins such as the status of disulfide bond between Cys residues and the sulfenylation as you mentioned. On the other hand, as a kind of toxic molecules, H₂O₂ can result in oxidative damage, so to scavenge over-accumulated H₂O₂ (ROS) is necessary for the survival under drought and growth recovery at post-drought period. Thus, it has been reported that the mutation / knock-down of both ROS producers (e.g., rice and Arabidopsis RBOHB mutation, Shi et al, Plant Cell Rep, 2020; He et al, BBRC, 2017) and ROS scavengers (e.g., wheat catalase knock-down) reduced drought tolerance of plants. Based on these, to fine-tune ROS homeostasis through orchestrating the machineries of both ROS production and scavenging is helpful for enhancing the tolerance to drought and other abiotic stresses. In this study, the overexpression and knock-down of *TaWD40-4B.1* did not affect catalase activity and H₂O₂ level under the well-watered condition but promoted the activity and alleviates the drastic increase of H₂O₂ level under drought, which may account for that *TaWD40-4B.1* does not lead to the trade-offs between grain yield and drought tolerance. We have updated this information in the Introduction and Discussion of the revised manuscript (Lines 85-91 in Introduction and lines 619-633 in Discussion).

4. Figures S10 and 3 show that *TaWD40-4B.1* (C or T) is localized in cytosol while TaCAT in peroxisomes. However, the interaction of *TaWD40-4B.1* with TaCAT

occurs both in the cytosol and peroxisomes. How to explain such results? In addition, the Figure 3B (BiFC) cannot rule out the possibility that the small puncta is other organelles or structures other than peroxisomes as no peroxisomal marker was used.

Answer: Thank you for your advice. The cellular localization of TaCATs and TaWD40-4B.1 as well as the place of the interaction between TaCATs and TaWD40-4B.1 has been re-analyzed by co-expressing peroxisome marker DsRed-SKL in wheat protoplasts (Fig 3B, S9A, S10). Moreover, we have confirmed the subcellular localization of TaCATs and TaWD40-4B.1 with the western blotting assay using the total, peroxisomal and cytoplasmatic proteins extracted from the protoplast (Fig S10B). Given that catalases majorly function in the peroxisomes, we speculate that the interaction of TaWD40-4B.1 with TaCATs around the peroxisomes may help TaCATs oligomerize and transport into the peroxisomes, which is an interesting question to be addressed in the future. The results have been added in the revised manuscript (Lines 271-284), and the methods had been updated and added in M&M (Lines 916-952).

5. Does TaWD40-4B.1 affect TaCATs expression?

Answer: Thank you for your advice. We have measured the expression of canonical catalase genes *TaCAT1* and *TaCAT3* in *TaWD40-4B.1* transgenic lines (The primers were designed to amplify the alleles of A, B and D subgenomes together), and found that TaWD40-4B.1 does not affect the expression of these genes (Fig S11). The results have been added in the revised manuscript (Lines 316-319).

6. The second leaves of the treated and untreated wheat seedlings were employed to analyze the H₂O₂ content, CAT activity and MDA content. What's the expression pattern of TaWD40-4B.1 in different tissues and organs of wheat?

Answer: Thank you for your advice. To avoid the influence of heterogeneity among different leaves, we selected the second leaves to measure these physiological indices. We have measured the expression of *TaWD40-4B.1* in different organs of wheat, and found that *TaWD40-4B.1* was expressed in all of these tissues at young and reproductive stages (Fig S13A). The results have been added in the revised manuscript (Lines 406-407), and the tissue sampling method for qPCR has been presented in M&M (Lines 726-728).

7. Does TaWD40-4B.1 has paralogs in A- or D-subgenome in wheat? If yes, are there natural variations in their gene structures, similar to the truncation found in TaWD40-4B.1C but not TaWD40-4B.1T?

Answer: Thank you for your advice. There have two paralogs (*TraesCS4A02G242800* and *TraesCS4D02G071100*) in A- and D-subgenomes in wheat. We sequenced the paralogs among the natural accessions, and found that there has no allelic variation in either the paralog of A- or D-subgenome. Moreover, we checked the public SNP databases of wheat accessions (please see the manuscript for the database information), and there also has no allelic variation in these two paralogs. This result

has been added in the Discussion (Lines 636-638).

8.Line 256-265 “Given that” was used two times in one paragraph. Please consider to change one of them as “Considering that” or “Because” etc.

Answer: Thank you for your advice. The second “Given that” has been changed as “Because” (Line 295).

9.Line 275 and 277 “BMP-TaCAT3A” should be “MBP-TaCAT3A”?

Answer: Thank you for your careful review. It has been corrected in the revised manuscript (Lines 305-307).

10.Authors should provide the category numbers for the antibodies, kit and important reagents.

Answer: Thank you for your advice. The category numbers for the antibodies, kits and some chemicals have been added in the M&M.

11.Figure 4C the “H2O2” in the y axis should be lowercase.

Answer: Thank you for your careful review. It has been corrected in the revised manuscript.

12.Figure S12A, “Wellwithheld” is wrong.

Answer: Thank you for your careful review. It has been corrected in the revised manuscript.

13.Figure S12K, the “mmol H2O2” in the y-axis is wrong. Please carefully check all the labeling in the figures.

Answer: Thank you for your careful review. It has been corrected. All the labeling in figures have been checked.

Reviewer #3 (Remarks to the Author):

This manuscript describes an interesting characterisation of the role of TaWD40-4B.1 in drought tolerance in wheat. GWAS analysis revealed that drought tolerance was associated with the qDT4B locus on chromosome 4BS. Drought tolerance is associated with the WD40 gene TaWD40- 4B.1. Wheat accessions carrying the truncated allele TaWD40-4B.1T have a lower drought tolerance than those with complete allele TaWD40-4B.1C. Modifications in TaWD40-4B.1C expression had no effect on plant growth and grain yield under water-replete conditions. Data are presented showing that the allelic nonsense variant of TaWD40-4B.1, produces a truncated peptide TaWD40-4B.1T, which is associated with the lower drought tolerance in wheat accessions. In contrast, the complete allele TaWD40-4B.1C is associated with enhanced drought tolerance and higher grain yields under drought stress. Crucially, TaWD40-4B.1C (but not TaWD40-4B.1T) interacts with the catalase proteins enhancing in vivo catalase activities and the tolerance to oxidative stress

under drought. WD40 proteins function as scaffolds for protein-protein interactions. In the present study, TaWD40-4B.1C is suggested to bind catalase in a manner that enhances drought tolerance by increasing the catalytic activity of the enzyme. However, this study provides little indication of the mechanisms whereby the protein encoded by TaWD40-4B.1C can interact with this peroxisomal iron-containing homotetrameric protein to increase the dismutation of H₂O₂ into H₂O and O₂. A mechanistic study/explanation of where the TaWD40-4B.1C protein/catalase interaction takes place and how this interaction modifies catalase activity is essential in order to advance current knowledge. The catalase proteins in plants are known to undergo a number of post-translational modifications and numerous catalase binding partners have been reported in the literature. The present manuscript ignores the extensive literature on catalase binding proteins that modify biotic and abiotic stress tolerance traits.

Answer: Thank you for your advice. The mechanism modulating catalase activity has been widely investigated in animal cells, the catalase activity can be affected by posttranslational modification such as phosphorylation and ubiquitination, binding of heme, and formation of oligomers. The formation of oligomers has proved to be the most common mechanism for activating catalase activity; oligomeric (majorly tetrameric) catalase has high activity, but monomeric form has lower activity. In plant cells, a set of proteins interacting with catalase have been identified, but only several proteins were studied regarding the mechanisms for modulating catalase activity: phosphorylation (OsSTRK1, AtCPK8), ubiquitination (OsAPIP6), maintaining optical conformation by chaperone AtNCA1, promoting abundance by OsSRL10. We found TaWD40-4B.1^C promoted catalase activity in the *in vitro* enzymatic assay system that did not contain heme and lacked the condition for protein modification, which suggests that TaWD40-4B.1 may promote catalase activity via influencing oligomer formation but not posttranslational modification of catalase or affecting heme binding. Indeed, we found that TaWD40-4B.1^C but not TaWD40-4B.1^T promoted the interaction between TaCAT and TaCAT, and the formation of oligomeric catalase holoenzymes *in vivo* and *in vitro* (Fig 4F-I), which is consistent with the previous study that rice catalase can form dodecameric holoenzyme containing 12 units with high activity and fast turnover rate of H₂O₂ (Zhang, et al. Molecular Plant, 2016). Thus, TaWD40-4B.1^C promoted the formation of oligomeric (dodecameric) catalase holoenzymes in wheat under both well-watered and drought conditions, suggesting the enhancement of TaWD40-4B.1 to catalase activity under drought condition is in relation to the oligomer (dodecamer) formation. The results and discussion have been added in the revised manuscript (Lines 351-377 in Results and Lines 593-610 in Discussion).

The extensive literatures on catalase binding proteins that modify biotic and abiotic stress tolerance traits have been added (Lines 94-102).

The place of the interaction between TaCATs and TaWD40-4B.1 has been re-analyzed by co-expressing peroxisome marker DsRed-SKL in wheat protoplasts (Fig. 3B, S9A).

Given that catalases majorly function in the peroxisomes, we speculate that the interaction of TaWD40-4B.1 with TaCATs around the peroxisomes may help TaCATs oligomerize and transport into the peroxisomes, which is an interesting question to be addressed in the future. The results have been added in the revised manuscript (Lines 271-284), and the methods have been updated and added in M&M (Lines 916-952).

I am concerned about the physiological relevance of some of the studies. Wheat seedlings were grown under well-watered conditions for two weeks and then subjected to drought by withholding water for 14 days. This is a very severe stress that will take the plants almost to death. At this point, the seedlings were re-watered and various shoot parameters were measured after 3 days. This study does not therefore examine the role of TaWD40-4B.1C in mild drought conditions but rather the ability to withstand drought-induced death.

Answer: Thank you for your advice. We conducted the phenotype analysis of wheat seedlings under the mild drought stress by water-withholding for seven days, and found that the difference among the transgenic lines under mild drought was not different as obvious as that under strong drought. According to our observation, the leaves of all lines became slightly wilted under mild drought that is difficult to compare the difference among the lines, and the slightly wilted leaves could erect quickly after re-watering. Thus, we did not present the result of mild drought stress.

Crucially also, the treatment with hydrogen peroxide involved culturing seedlings at the three-leaf stage with 100mM hydrogen peroxide for one week. This is an excessive amount of oxidant that would most certainly kill the seedlings.

Answer: Thank you for your advice. The concentration (100 mM) of H₂O₂ was selected according to the references that phenotyped the tolerance of wheat seedlings to oxidative stress using exogenous H₂O₂ treatment (Liu et al, Plant Cell, 2014; Qiu et al, Plant Biotechnology Journal, 2021). Besides, 100 mM H₂O₂ was also used for measuring the oxidative stress tolerance of maize (Qin et al, Plant Journal, 2021) and rice seedlings (Zhou et al, Plant Cell, 2018; Ni et al, Journal of Integrative Plant Biology, 2022). We repeated the phenotyping assay with 100 mM H₂O₂ several times, and the difference in the growth of seedlings were reproducible. This assay was used to compare the difference in the tolerance to oxidative stress among the lines; because 100 mM treatment can differentiate the difference, we did not conduct the assay using other concentrations of H₂O₂.

The levels of hydrogen peroxide shown in Figure 5C are given as micromoles per gm fresh weight. The values are far too high. Values should be in nanomoles per gm fresh weight. I am therefore very concerned about the methods used for the extraction and assay of hydrogen peroxide, which is notoriously difficult to extract and assay without artefact (see for example: Noctor et al. (2016) Oxidative stress and antioxidative systems: recipes for successful data collection and interpretation. Plant Cell Environment. 39. 1140-1160. Since the effects of TaWD40-4B.1C on the ability of catalase to prevent accumulation of hydrogen peroxide, accurate data on this

metabolite are essential.

Answer: Thank you for your advice. As mentioned in M&M, wheat leaves were snap-frozen in liquid nitrogen for measuring H₂O₂ content using the H₂O₂ Colorimetric Assay Kit protocol (Beyotime) according to the manual. H₂O₂ content was also calculated according to the method in the manual. The measurement was repeated for three times before the original submission. During the revision of this manuscript, we repeated the measurement again, and found that the contents were also in the range of micromoles per gm fresh weight. The H₂O₂ contents among the samples in the genetic analysis were also measured as micromoles per gm fresh weight (Fig 6F). Besides, the H₂O₂ contents were measured as micromoles per gm fresh weight in some reports in the leaves of other crops such as maize, durum wheat, rice and cassava (Qin et al, Plant Journal, 2021; Feki et al, Plant Physiology and Biochemistry, 2015; Yan et al, Plant Journal, 2021; Wang et al, Plant Biotechnology Journal, 2023). As said in the review you mentioned, there have many challenges of measuring ROS in plants. In our work, the difference in H₂O₂ contents among cultivars SR3/YM20 and their transgenic lines were repeated for many times, in line with the role of TaWD40-4B.1 in the *in vitro* and *in vivo* catalase activities, we could conclude that TaWD40-4B.1 can modulate H₂O₂ contents in wheat upon drought stress.

The authors state that the relative quantification of hydrogen peroxide was based on 3,3-diaminobenzidine (DAB) staining but this method is qualitative and not quantitative. To make this assay quantitative, internal standards of known hydrogen peroxide concentration are required, and this is not possible.

Answer: Thank you for your advice. DAB staining is an easy and convenient method for measuring H₂O₂ content, and has been often used for comparing H₂O₂ levels in plants (for example, Xiong et al, PNAS, 2021). As you said, DAB staining can't accurately measure H₂O₂ level, because of its qualitative characteristics. Thus, besides DAB staining, we also measured the H₂O₂ level using a quantitative method with H₂O₂ content measurement kit, and the results of two methods are consistent (Figs 5, S12). DAB staining provides further evidence that TaWD40-4B.1 can modulate H₂O₂ contents.

Reviewer #4 (Remarks to the Author):

Dear Authors,

Your study to investigate the influence of an allelic variation inside the TaWD40-1 4B.1 and how it can contribute to drought tolerance is a very impressive and comprehensive study.

I have a few comments to point out

why did you use 198 wheat accessions only?

Answer: To gain accurate phenotyping data during the water-withholding assay is very crucial for GWAS. As we all know, the phenotype upon drought is susceptible to micro-environmental conditions during the water-withholding assay, so a large population will enlarge the effect of micro-environment and make the phenotyping data difficult to get accurate genetic dissection on drought tolerance via GWAS. Thus, in this work, 210 accessions were selected for water-withholding assay, and each accession had five replicates via the completely randomized design to ensure the accuracy of phenotyping. The 210 accessions were selected from the natural population (more than 1000 accessions) used for analyzing the genetic diversity of wheat germplasm from different wheat growth districts around the world, especially in China, according to the following principles: (1) the genetic and geographical diversities of selected accessions can represent those of the whole natural population; (2) The LD decay should not be obviously distinct from that of the whole natural population; (3) Outstanding accessions in the drought or semi-drought wheat area should not be completely omitted. Even so, 12 accessions were rejected because of low phenotype repeatability, and other 198 accessions were used for GWAS. According to the results of this work, the considerable phenotypic variation (Fig S2D; Table S1) and the genetic diversity (Fig S1) were similar to those using the accessions selected from the same natural population (Wu et al., 2020; Wu et al., 2021; Yu et al., 2020), showing these 198 accessions are qualified for GWAS.

In recent years, increasing studies used natural populations in controlled scales for GWAS in plants. For example, most studies based on GWAS in rice used several hundred and even nearly two thousand accessions, while the GWAS concerning the geographical adaptation to soil nitrogen in rice used 110 accessions (Liu et al., 2021), and the genetic and phenotype diversity of this small population was enough for conducting genetic analysis. Based on these reasons, we used 198 accessions for GWAS in this work.

Liu, Y., Wang, H., Jiang, Z., Wang, W., Xu, R., Wang, Q., Zhang, Z., Li, A., Liang, Y., Ou, S., et al. (2021). Genomic basis of geographical adaptation to soil nitrogen in rice. *Nature* 590, 600-605.

Wu, J., Wang, X., Chen, N., Yu, R., Yu, S., Wang, Q., Huang, S., Wang, H., Singh, R.P., Bhavani, S., et al. (2020). Association Analysis Identifies New Loci for Resistance to Chinese Yr26-Virulent Races of the Stripe Rust Pathogen in a Diverse Panel of Wheat Germplasm. *Plant Dis* 104, 1751-1762.

Wu, J., Yu, R., Wang, H., Zhou, C., Huang, S., Jiao, H., Yu, S., Nie, X., Wang, Q., and Liu, S. (2021). A large-scale genomic association analysis identifies the candidate causal genes conferring stripe rust resistance under multiple field environments. *Plant biotechnology journal* 19, 177-191.

Yu, S., Wu, J., Wang, M., Shi, W., Xia, G., Jia, J., Kang, Z., and Han, D. (2020). Haplotype variations in QTL for salt tolerance in Chinese wheat accessions identified by marker-based and pedigree-based kinship analyses. *The Crop Journal*.

have they geographically distributed over the world?

Answer: Thank you for your question. The geographical distribution of the 198 accessions is listed in Table S2. Among them, 140 accessions are from China, 54 accessions are from Europe, North America, South America and Asia, and the other four accessions are unknown.

did you select them based on different biological status, breeding line, cultivars, landraces and wild?

Answer: Thank you for your question. 1. As shown in Table S2, most of the accessions for GWAS are from China, and these accessions are planted in different main growing areas with different annual rainfalls. 2. The accessions from the other counties are majorly from the wheat major growing areas around the world. 3. We also considered the types of accessions, and 198 accessions consist of seven breeding lines and 13 landraces.

how many SNP markers did you use in GWAS at the end?

Answer: Thank you for your advice. In total 419,606 high-quality SNP markers are retained after filtering. The data has been present in Line 140-143.

it was not clear, how the GWAS strategy had been done, please find a reference for GWAS in cereals.

Answer: Thank you for your advice. The detailed description of GWAS strategy has been improved (Lines 697-722), and the reference has been added (Line 707).

it was not clear how did you reach the specific allele.

did you resequence the gene in the accessions?

Answer: Thank you for your question. Yes, we sequenced this gene using 198 accessions by the Sanger DNA sequencing method, and found that there have four SNPs in the gene. These four SNPs are completely linked to produce two haplotypes. The results have been presented in the revised manuscript (Lines 180-183).

how can you elucidate your conclusion that *TaWD40-4B.1* was selected by breeding? Please make sure to provide proof of that assumption.

Answer: Thank you for your question. Firstly, the proportion of *TaWD40-4B.1^T* in cultivated hexaploid wheat accessions is higher than the proportion in the landraces (Figs 7B, S17). Thus, *TaWD40-4B.1^T* appeared to be selected during the breeding of wheat, partially because this gene (the drought tolerance QTL in 4B) is close to grain height gene *Rht* and grain yield. Secondly, the proportion of *TaWD40-4B.1^T* in the accessions from the areas with less annual rainfall is lower than that from the areas with more annual rainfall (Fig 7C, D). Based on these, we speculated that *TaWD40-4B.1* was selected during breeding. The comprehensive analysis of the evolution characteristics of this gene as well as the association between the allelic variation and the breeding selection needs to be performed in the future. According to your advice, we have revised the presentation regarding this speculation (Lines 36-38 in Abstract, lines 647-662).

can you show the sequence differences between Yangmai 20 (YM20) carrying the 189 TaWD40-4B.1T allele and Shanrong 3 (SR3) carrying the TaWD40-4B.1C.

Answer: Thank you for your question. YM20 harbors the truncated haplotype, and SR3 harbors the complete haplotype, which has been present in Line 207-210.

can you present the gene construct of RNAi.

Answer: Thank you for your advice. The construct of RNAi has been improved (Line 744-749).

can you provide the ANOVA or any other statistical analysis to confirm the results of grain yield-related traits under both drought and control conditions?

Answer: Thank you for your advice. The data of grain yield and yield-related traits have been analyzed by the one-way ANOVA, and the post-hoc comparison of ANOVA has been analyzed by the Tukey method. The results have been provided in the Figs 7 and S16, and the statistical methods have been added in the figure legends. Besides, we have added the F and t statistics, degrees of freedom and *P* values in the figures.

Reviewers' Comments:

Reviewer #1:

Remarks to the Author:

The authors almost answer all of my question. Although I still have some concerns on the catalase activity and the genetic linkage of two genes, considering this is a study using wheat as a material, it is already a very good study.

Reviewer #2:

Remarks to the Author:

I am happy with the revisions and have no further comments.

Reviewer #3:

Remarks to the Author:

This manuscript reports a novel and interesting study that identified a protein that is important in drought tolerance in wheat and demonstrated that it interacts with canonical catalases, promoting oligomerization and increasing activity. This is a significant new finding that merits publication. However, the authors have not addressed my concerns regarding the very high levels of hydrogen peroxide that were measured in the wheat leaves. Clearly, many techniques for the measurement of hydrogen peroxide are flawed and prone to artefact. I recommend that the hydrogen peroxide data are removed from this manuscript prior to publication because these data are not essential for data interpretation. Indeed, much of the hydrogen peroxide measured is likely to be localised in the apoplast/cell wall compartment rather than in the cytoplasm.

Reviewer #4:

Remarks to the Author:

Dear Authors

thanks for answering my comments and including my suggestion in the manuscript.

I'm satisfied with the current version of the manuscript

Response to the comments of four reviewers

Reviewer #1 (Remarks to the Author):

The authors almost answer all of my question. Although I still have some concerns on the catalase activity and the genetic linkage of two genes, considering this is a study using wheat as a material, it is already a very good study.

Answer: Thank you for your comments.

Reviewer #2 (Remarks to the Author):

I am happy with the revisions and have no further comments.

Answer: Thank you for your comments.

Reviewer #3 (Remarks to the Author):

This manuscript reports a novel and interesting study that identified a protein that is important in drought tolerance in wheat and demonstrated that it interacts with canonical catalases, promoting oligomerization and increasing activity. This is a significant new finding that merits publication. However, the authors have not addressed my concerns regarding the very high levels of hydrogen peroxide that were measured in the wheat leaves. Clearly, many techniques for the measurement of hydrogen peroxide are flawed and prone to artefact. I recommend that the hydrogen peroxide data are removed from this manuscript prior to publication because these data are not essential for data interpretation. Indeed, much of the hydrogen peroxide measured is likely to be localised in the apoplast/cell wall compartment rather than in the cytoplasm.

Answer: Thank you for your comment. We deleted the data of H₂O₂ levels in Figures 5, 6 and S12. Accordingly, the panel order and legends of these figures have been updated, and the sentences in Results, Discussion and Methods have been deleted.

Reviewer #4 (Remarks to the Author):

Dear Authors

thanks for answering my comments and including my suggestion in the manuscript.

I'm satisfied with the current version of the manuscript

Answer: Thank you for your comments.